# Tracing immune cells around biomaterials with spatial anchors during large-scale wound regeneration

Yang Yang [1,2,3], Chenyu Chu [1,3], Li Liu[1,3], Chenbing Wang[1], Chen Hu[1], Shengan Rung[1,2], Yi Man [1,4] ✉ & Yili Qu [2,4] ✉

Skin scarring devoid of dermal appendages after severe trauma has unfavorable effects on aesthetic and physiological functions. Here we present a method for large-area wound regeneration using biodegradable aligned extracellular matrix scaffolds. We show that the implantation of these scaffolds accelerates wound coverage and enhances hair follicle neogenesis. We perform multimodal analysis, in combination with single-cell RNA sequencing and spatial transcriptomics, to explore the immune responses around biomaterials, highlighting the potential role of regulatory T cells in mitigating tissue fibrous by suppressing excessive type 2 inflammation. We find that immunodeficient mice lacking mature T lymphocytes show the typical characteristic of tissue fibrous driven by type 2 macrophage inflammation, validating the potential therapeutic effect of the adaptive immune system activated by biomaterials. These findings contribute to our understanding of the coordination of immune systems in wound regeneration and facilitate the design of immunoregulatory biomaterials in the future.

After severe skin damage, the resulting scar usually contains dense extracellular matrix (ECM) fibers devoid of the hair follicle (HF) and sebaceous gland (SG), which lack sensation and endocrine function as well as the flexibility of normal skin[1]. As a result, there is an urgent need to explore the fundamental mechanisms stimulating HF regeneration in skin repair. The immune system plays a varying role in driving scar fibrosis[2–4] or HF regeneration[5–8] upon different environmental stimuli. Macrophages contribute to all phases of tissue repair, and the heterogeneity of macrophages is believed to be one of the critical orchestrators determining the healing outcome[9,10]. Two major subpopulations, including pro-inflammatory M1-like and anti-inflammatory M2-like, have been credited with these distinct roles[11]. It has been reported that the pro-inflammatory macrophage-elicited pro-inflammatory mediators, such as tumor necrosis factor (TNF)[5] and interleukin-1 beta (IL-1β)[12], effectively promote subsequent HF

neogenesis. In contrast, type-2 anti-inflammatory macrophages might play an essential role in wound fibrosis by promoting fibrotic fibroblast activation and collagen cross-linking[3] via fibrotic cytokines such as transforming growth factor–beta (TGF-β)[13] and RELMα[14] or chronic phagocytosis activity[4] at a later stage. Although the role of macrophages in pathogen clearance and tissue fibrosis[15] has long been stressed, only recently have the T lymphocytes been more thoroughly investigated. In addition to γδ T cells that induce HF neogenesis through the secretion of fibroblast growth factor 9 (Fgf9)[6], the adaptive T cells are being explored to gain more understanding of its role in regulating macrophage polarization and, thus, wound regeneration[2]. T cells coordinate the polarized immune responses through differentiation into specialized subsets of helper T cells (Th1, Th2, and Th17) and drive the type 1/2/3 paradigm of immunity[16]. Besides, specialized regulatory T cells (Tregs) have evolved to counterbalance the

[1]Department of Oral Implantology & State Key Laboratory of Oral Diseases and National Clinical Research Center for Oral Diseases, West China Hospital of Stomatology, Sichuan University, Chengdu 610041, China. [2]Department of Prosthodontics & State Key Laboratory of Oral Diseases and National Clinical Research Center for Oral Diseases, West China Hospital of Stomatology, Sichuan University, Chengdu 610041, China. [3]These authors contributed equally: Yang Yang, Chenyu Chu, Li Liu. [4]These authors jointly supervised this work: Yi Man, Yili Qu. ✉e-mail: manyi780203@126.com; qqyili@126.com

potentially detrimental effect of the innate immune system by suppressing macrophage response[17] and facilitating wound regeneration.

Recently, tissue regeneration mediated by immunoregulatory biomaterials are emerging as a prospective strategy in tissue engineering[18,19]. Biological cues can be integrated into a polymer scaffold to mimic the native ECM, which guides tissue regeneration[20,21]. Upon materials implantation, the foreign body responses (FBR) process initiates with an immune response. It has been shown that modulating FBR by adjusting biomaterials characteristics may create a desired biological response to mobilize stem cells or stimulate specific cell proliferation[20,21]. Our previous studies had reported the Aligned nanofibers scaffold with an immunomodulatory effect in accelerating small skin wound (diameter = 6 mm) re-epithelialization[22,23]. However, scar tissue induced by a 6-mm-scale wound was tiny (diameter = 1～1.5 mm). Considering the more obvious inflammatory response and larger detectable scar tissue, the large-scale full-thickness wound model (diameter = 10～20 mm)[24–28] provided a better media through which we can easily evaluate the pro-regenerative effect of ECM scaffolds regarding scarless wound healing.

Currently, high-resolution techniques such as single-cell RNA sequencing (scRNA-seq)[22,29,30] have been applied to identify rare cell subpopulations in the implantation model but lack information on spatial distribution. Development in spatial transcriptomics (ST) has enabled the assessment of gene expression at spatial resolution[31], which has been applied to the study of cancer[32,33], liver[34], and brain tissue[35] to detect regional cellular communication. The wound healing model with implanted biomaterials provides an ideal method to understand the FBR and probe the role of the immune system in tissue regeneration. To our knowledge, multi-omic approaches, including scRNA-seq and ST, were first applied to trace spatial heterogeneity in the biomaterial-mediated skin wound healing process in this study.

Here, we unveil the cell composition around the ECM scaffold in both wildtype and immunodeficient mice to anchor the critical role of T cells in HF regeneration, which would help optimize the existing biomaterial constructs and provide feasible strategies for the design of novel immunoregulatory products in tissue engineering for biomedical use.

## Results

### Enhanced HF neogenesis in scaffold-implanted wounds with activation of the adaptive immune system

The workflow for evaluating large wound healing is summarized in Fig. 1a. We placed ECM scaffolds below the large wound (diameter = 1.5 cm) in the ECM_LW (Large wound treated with ECM scaffold) group, while the Ctrl_LW (Large wound treated with saline) group received no biomaterials (Fig. 1b). C57BL/6 mice are a classical choice for studying wound healing due to their accessibility, affordability, and ease of handling. It is worth noting that wound healing in rodent models primarily relies on contraction by the panniculus carnosus for wound closure, whereas in humans, re-epithelialization and granulation tissue formation play a larger role[36]. As shown in Supplementary Fig. 1a, we created a splinted wound excisional model that could restrict the contraction of the panniculus carnosus, while the unsplinted group was treated without a silicone splint. The decreased rate of wound closure (Supplementary Fig. 1b) and the increased granulation formation and de novo HFs were observed in the splinted mice (Supplementary Fig. 1c–e). In order to minimize wound contraction in rodents and mimic the wound-healing process in humans with tight skin, we chose the splinted wound-healing model for further evaluation. Wound coverage was faster in the ECM_LW group on postoperative day (POD) 7 (Fig. 1c, d), and the ECM_LW group also had a smaller epithelial gap width on POD 7 and POD 14 (Fig. 1e, f). Immunofluorescence (IF) staining for cytokeratin 5 (KRT5) and cytokeratin 10 (KRT10) showed that keratinocytes crawled around the biomaterials,

and the ECM_LW group owned a larger area of neo-epithelium (Supplementary Fig. 2a).

In the Ctrl_LW group, there was limited HF neogenesis restricted in the wound center since the POD 21. In comparison, we observed that the ECM_LW group recapitulated the normal skin architecture with an equivalent number of mature HF (Fig. 1e, h, Supplementary Fig. 1e). The nascent HFs induced by ECM scaffolds mimicked embryonic hair follicle development pattern (Fig. 1e, g), with high KRT17 (green) in hair germ (HG) and TWIST2 (red) in dermal condensate (Dc) on POD7. After morphogenesis, neogenic HFs in the ECM_LW group contained proliferating epithelial cells expressing Ki67 (red) with sebaceous glands (SCD1/ green). Of note, the ECM membrane implanted in the large wound did not trigger any obvious fibrous capsule and exhibited an appropriate degradation rate in vivo. We observed degrading fragments on POD14, and no visible particles remained on POD28 (Fig. 1e). The excellent biocompatibility and degradability of the membrane could prevent the risk of immune rejection and secondary surgery in further clinical applications.

To explore the underlying mechanism, a bulk-tissue RNA-seq (bulk-seq) analysis was conducted for two groups harvested on POD 7 (n = 3 for each group). Gene enrichment analysis of Ctrl_LW group up-regulated genes (p value < 0.05 and |log2FoldChange| > 1) illustrated a state readied to incite innate immune responses (Fig. 1i and Supplementary Fig. 2b), indicated by neutrophil chemotaxis and macrophage activation via type 2 immune response. It has been reported that type 2 cytokines such as interleukin 4 receptor, alpha (IL-4Rα) could activate anti-inflammatory macrophages and lead to the cross-linking of collagen fibers in scar formation[3]. The accumulation of type 2 myeloid immune cells in the Ctrl_LW group might be the reason for excessive extracellular matrix organization. In contrast, enrichment analysis of ECM up-regulated genes (p value < 0.05 and |log2FoldChange| > 1) revealed enrichment of hair follicle development and mesenchyme morphogenesis driven by Wnt signaling pathway and Hedgehog signaling pathway in the ECM_LW group (Fig. 1i and Supplementary Fig. 2c). The role of Wnt and Hedgehog signaling pathway in regulating T cell development[37–39] as well as hair follicle regeneration[40,41] had been stressed. We noticed that genes such as frizzled class receptor 5 (Fzd5), GATA binding protein 3 (Gata3), and GLI family zinc finger 3 (Gli3) also enriched in T cell differentiation in the thymus. Besides, Gata3 expressed by regulatory T cells (Tregs) are proven to be necessary to prevent excessive collagen deposition driven by type-2 macrophage inflammation[42], which implicated the importance of adaptive immune system homeostasis in material-primed skin regeneration. Flow cytometry (Fig. 1j and Supplementary Fig. 3) and immunohistochemistry (IHC) staining (Supplementary Fig. 2d) confirmed the increased T cell (CD3+) infiltration around implanted biomaterials in the early phase.

### A single-cell atlas of the biomaterials tissue microenvironment

To explore the spatial characteristics of single cells during wound healing, we applied ST and scRNA-seq to compare the spatial gene expression profiles between two groups (Fig. 2a). At first, to explore the cell composition in the biomaterial-treated wound, we isolated cells from the ECM_LW and Ctrl_LW samples on POD 7, 14, and 21, and applied them to the 10x scRNA-seq platform (Supplementary Fig. 4a). After cell filtering, unsupervised clustering of Seurat categorized the cells into clusters based on global gene expression patterns. Later, clusters were then assigned to first-level main classes of cells. The composition of each main cluster was listed so that the proportion of cells from two groups could be identified across all cell clusters (Supplementary Fig. 4b). Marker genes for each main cluster were shown in the heatmap and listed in Supplementary Fig. 4c. Volcano plot showed genes related to the innate immune system were up-regulated in the Ctrl_LW group (Supplementary Fig. 4d).

Firstly, we selected main clusters defined as keratinocytes and subjected them to a second round of unsupervised clustering

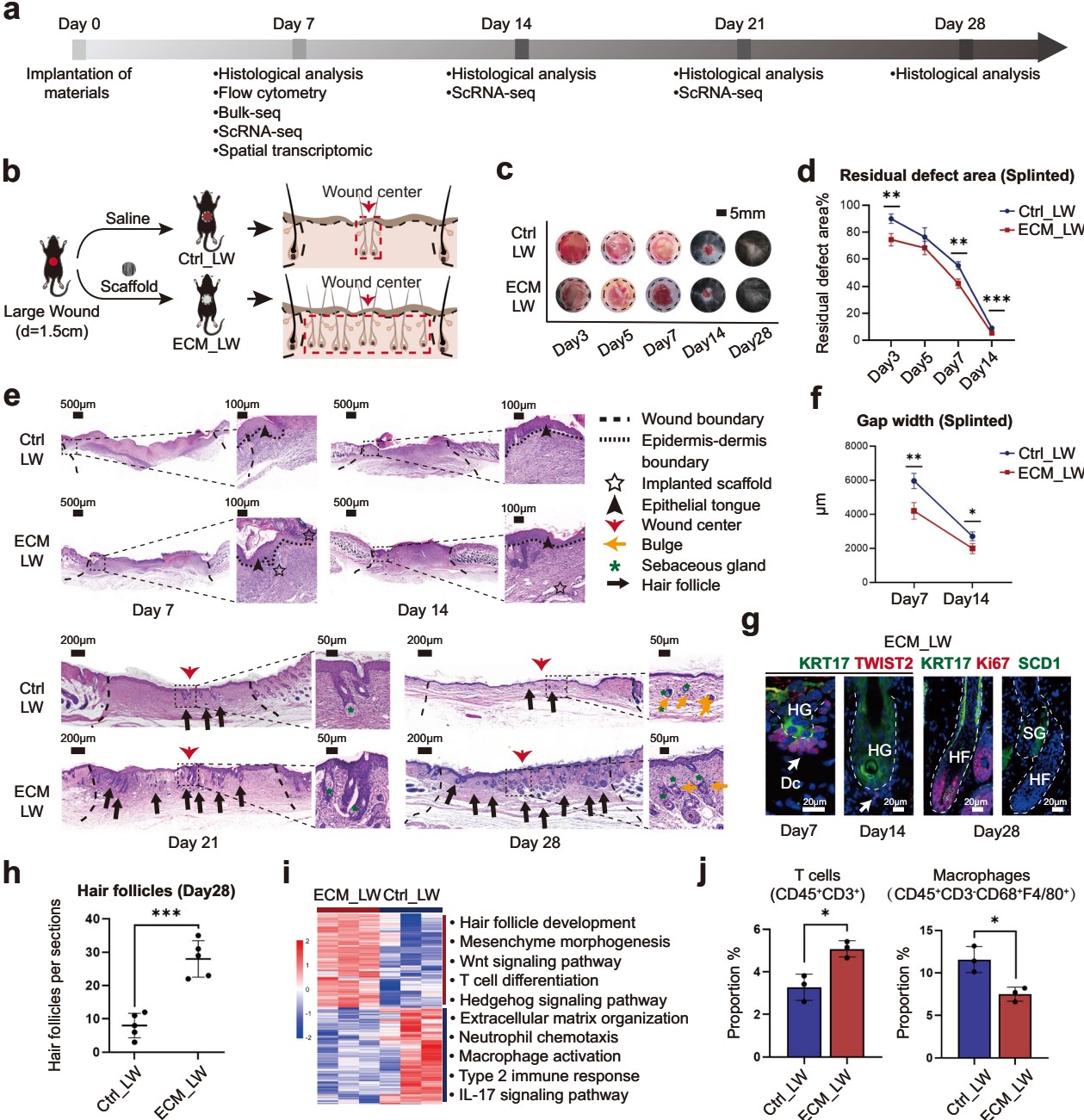

**Fig. 1 | Evaluation of the wound healing process treated with ECM scaffolds.**
**a** Workflow for evaluating large-scale wound healing. **b** Surgical processes for skin splinted excisional wound model. **c** Residual wound area at 3, 5, 7, 14, and 28 days, black dashed circles denoting the original wound area. **d** corresponding analysis of residual wound (Data are presented as mean ± SD, n = 4 biologically independent samples, two-tailed t-test, Day3 **p = 0.002; Day5 p = 0.104; Day7 **p = 0.001; Day14 ***p = 0.000444). **e** Representative H&E images of two groups at 7, 14, 21, and 28 days. **f** Quantitative evaluation of the gap width of neo-epithelium (Data are presented as mean ± SD, n = 4 biologically independent samples, two-tailed t-test, Day7 **p = 0.002; Day14 *p = 0.011). **g** Representative IF images of nascent HFs within the ECM_LW group, stained for KRT17 (green) and TWIST2 (red), Ki67 (red), and SCD1 (green), respectively. Abbreviations: HF, hair follicle; HG, hair germ; Dc, dermal condensate; SG, sebaceous gland. **h** Histologic quantification of de novo HFs on POD28 (Data are presented as mean ± SD, n = 5 biologically independent samples, two-tailed t-test, ***p = 0.000141). **i** Bulk-RNA sequencing analysis of ECM_LW versus Ctrl_LW mice on POD7 (n = 3 for each group). Heatmap (left) showing hierarchical clustering of differentially expressed genes (p value < 0.05 & | log2FC | > 1) between two groups, and corresponding gene set enrichment analysis (right) showing the enriched terms in ECM_LW (top) versus Ctrl_LW (bottom) groups. **j** Proportions of T cells (CD45+CD3+) and macrophages (CD45+CD3-F4/80+CD68+) cell populations in the wound environment on POD7, determined by flow cytometry (% = the number of target cells / the number of all single live cells) (Data are presented as mean ± SD, n = 3 biologically independent samples, two-tailed t-test, T cells *p = 0.013; Macrophages *p = 0.016). p value: *p < 0.05, **p < 0.01, ***p < 0.001, and ****p < 0.0001.

(Fig. 2b). The heterogeneity of keratinocyte subclusters of this dataset corresponded with the healing outcomes of two groups: the higher proportion of Krt5+ interfollicular epidermal basal cell[1] (IFEB[1]) and Krt10+ interfollicular epidermal differentiated cell[1] (IFED[1]) were observed in the ECM_LW group on POD7, supporting more neo-epithelium proliferation in the presence of scaffolds in the proliferative stage (Supplementary Fig. 2a). HF neogenesis was believed to occur through the migration of epithelial HFSC or hair

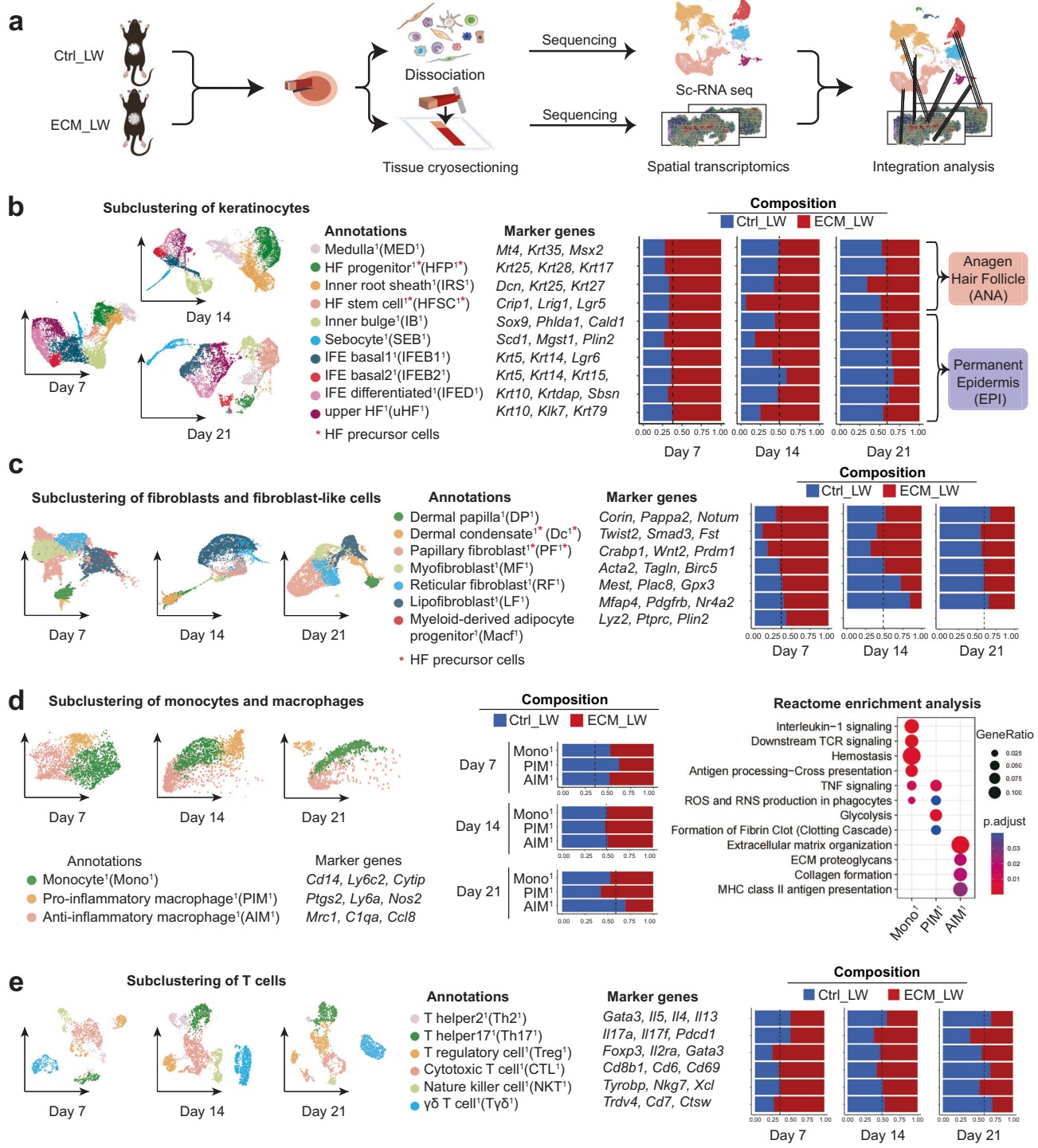

**Fig. 2 | The single-cell atlas of the biomaterials-mediated microenvironment.**
**a** Schematic for generating scRNA-seq and spatial transcriptomics data from large area excisional wounds on POD 7, 14, and 21. **b** Subclustering of keratinocytes showing four subsets from the anagen hair follicle and six subsets from the permanent epidermis. The composition and marker genes for each subset are listed. **c** Subclustering of fibroblasts showing two fibroblast-like subsets and five fibroblast subsets. The marker genes and composition for each subset are listed.
**d** Subclustering of monocyte/macrophage showing three subsets. The marker genes, composition, and enrichment analysis for each subset are listed.
**e** Subclustering of T cells showing six subsets. The marker genes and composition for each subset are listed.

follicle progenitor (HFP) to the wound center and form the placodes to activate papillary fibroblast (PF) fate specification into dermal condensate (Dc)[40,43,44]. In accordance with reports, we found a higher proportion of *Krt25+Krt28+Krt17+* hair follicle progenitor[1]

(HFP[1])[22] and *Crip1+Lrig1+Lgr5+* hair follicle stem cell[1] (HFSC[1])[5] in ECM_LW group on POD7, 14 and 21 (Fig. 2b), which might serve enough epithelial resources for the following nascent HFs reconstruction.

Fibroblasts are the major mesenchymal cells in the dermal layer of skin, and different fibroblast subclusters occupy distinct locations exhibiting considerable functional diversity. In general, dermis fibroblasts arise from two distinct lineages:[19,45] (1) upper lineage: papillary fibroblast (PF), which contacts the epidermis and gives rise to the dermal component of HFs; (2) lower lineage: reticular fibroblast (RF) which synthesize most of the ECM protein and lipo-fibroblast (LF) which provides the preadipocytes progenitors of the hypodermis. Besides, skin also contains specialized fibroblast-like cells, including dermal papilla (DP) and dermal condensate (Dc), which have unique transcriptional characteristics with universal fibroblast[46]. DP is located at the base of mature hair follicles and serves as the principal signaling niche of hair follicle activities. Origin from PF, dermal condensate (Dc) is believed to be the progenitor of DP in embryonic development[44,47]. In this dataset, we defined five fibroblasts and two fibroblast-like cells subclusters based on defined markers published before[22,43,48,49] (Fig. 2c). All subclusters expressed pan-fibroblast marker platelet-derived growth factor receptor-a (*Pdgfra*)[43], while fibroblast-like cells showed the lower expression level of dermatopontin (*Dpt*)[50] and higher expression of pappalysin 2 (*Pappa2*)[46] (Supplementary Fig. 5a). In accordance with previous report[43,51], the primary wave of dermal repair in the Ctrl_LW group was mediated by the lower lineage fibroblast including *Gpx3*+*Mest*+ reticular fibroblast[1] (RF[1]) and *Mfap4*+ *Cd34*+ lipo-fibroblast[1] (LF[1]) on POD7 and POD14, which was respectively related to dermis collagen fibril organization and hypodermis adipocytes formation in GO enrichment analysis (Supplementary Fig. 5b). It's worth noting that RF[1] highly expressed genes enriched in innate immune system, in which *Il33*[52], *I4ra*[14] and *Il13ra1*[9] were related to the initiation of type 2 macrophage inflammation and collagen deposition in fibrous disease. In contrast, there were more upper lineage *Crabp1*+*Prdm1*+ PF[1] in biomaterial-implanted wounds, which was believed to have the capacity to support HF initiation[48,53] (Supplementary Fig. 5b). Remarkably, we identified the *Twist2*+*Smad3*+ dermal condensate[1] (Dc[1]) cell[44] in this dataset, which was verified by histology (Fig. 1e) and IF (Fig. 1g). In embryologic HF morphology, Dc is acting as the signaling niches to stimulate epithelial placode growth[44], and thus promote HF morphogenesis. Since there is a significantly higher proportion of Dc[1] and PF[1] in the ECM_LW group on POD 7 and POD14 (Fig. 2c), which might provide enough mesenchymal component for the subsequent HF formation in the biomaterial-implanted group too.

For the immune micro-environment, most myeloid cells owned the lower proportion in the ECM_LW group on POD7 (Supplementary Fig. 4b). Firstly, as the first responding immune cells acting as the critical mediators of the innate immune system, neutrophils are necessary for the recruitment and differentiation of monocytes[54], which were peaked at the early stage. Most neutrophil subclusters owned a higher proportion in the Ctrl_LW group on POD7 and showed gene enrichment in pro-inflammatory pathways (Supplementary Fig. 6a). There was no noticeable difference in the proportion of dendritic cell (DC) between the two groups on POD7, 14, and 21 (Supplementary Fig. 4b). DC was classified into two subclusters, including *Krt5*+*Krt14*+ Langerhans cell[1] (LC[1]) and *Cd86*+ monocyte-derived dendritic cell[1] (MDC[1]), respectively related to phagosome and antigen presentation. There was no obvious difference in the proportion of DC subsets between the two groups (Supplementary Fig. 6b).

Monocytes and macrophages (MAC) were vital in biomaterial-related FBR[55], orchestrating tissue repair by modulating fibroblast activation[9]. Three subsets of MAC, including *Cd14*+*Ly6c2*+ monocytes[1] (Mono[1]), *Ptgs2*+*Nos2*+ pro-inflammatory macrophages[1] (PIM[1]), and *Mrc1*+ *C1qa*+ anti-inflammatory macrophages[1] (AIM[1]), were determined (Fig. 2d). As shown in enrichment analysis, pro-inflammatory (classically /M1-like) macrophages can be activated from monocyte by a variety of alarmins, which result in the expression of pro-inflammatory cytokines including TNF and IL-1β. Induced by the IL-4 and IL-13, anti-inflammatory (alternatively activated /M2-like) subsets are the foremost effectors of type 2 immune response, which play an essential role in aberrant collagen deposition[14] through the expression of TGF-β and various matrix remodeling mediators[56]. Of note, AIM[1], which enriched genes related to collagen formation, was the only subcluster that continuously decreased in the ECM_LW group on POD 7, 14, and 21, suggesting that the scaffold might play a role in reducing type 2 macrophage infiltration.

Although T cells are not required for wound healing, they are capable of modulating each phase of tissue repair by regulating macrophage activity. T cells were the only immune cell that owned a higher proportion in the ECM_LW group (Supplementary Fig. 4b), and genes related to T cells were substantially higher expressed by the ECM_LW group (Supplementary Fig. 4d), suggesting the activation of adaptive immune system induced by biomaterials. Subclustering of T cells resulted in six subsets including *Cd8b1*+ cytotoxic T cell[1] (CTL[1]), *Tyrobp*+*Nkg7*+*Xcl*+ nature killer cell[1] (NKT[1]), *Il17a*+*Il17f*+ T helper17[1] (Th17[1]), *Foxp3*+*Gata3*+ T regulatory cell[1] (Treg[1]), *Trdv4*+*Cd7*+ γδT cell[1] (Tγδ[1]) and *Gata3*+*Il4*+ T helper2[1] (Th2[1]) based on markers from published research[57,58]. More cells in the Th2[1] populations were from the Ctrl_LW samples on POD7, 14, and 21, whereas the larger number of Treg[1] were from the ECM_LW samples. *Il4*+ *Il13*+ Th2 cells are one of the effector cells associated with type 2 immune responses, which could facilitate the differentiation of type-2 AIM[52]. Located in the dermis near HF in normal skin, regulatory T cells (Tregs) are believed to maintain the balance of immune homeostasis. It has been reported that *Gata3*+ Tregs could restrain Th2-mediated fibroblast activation and scarring in murine cutaneous fibrosis[42], and clinical IL-2 therapy (which facilitates Treg proliferation and activation) has also been proven to be effective in alleviating chronic skin fibrosis[59]. In addition, Tregs could also facilitate HFSC differentiation to initiate HF regeneration through Jag1-Notch1 pathways[60]. Gene enrichment analysis confirmed that the Treg[1] subpopulation was enriched in signaling pathways regulating pluripotency of stem cells (Supplementary Fig. 5c), suggesting that Treg[1] recruited by ECM scaffolds might initiate HF neogenesis by both immune-suppression and pro-reparative functions[61].

### Discovery of spatial heterogeneity around biomaterials

To explore the spatial characteristics of cell heterogeneity in scaffold-implanted wounds, we applied ST to analyze the spatial gene expression profiles. Anatomical structures were identified in the ST sections (Fig. 3a). Based on the gene enrichment analysis of ST (Fig. 3b), the Ctrl_LW group enriched the genes related to extracellular matrix organization, collagen biosynthetic process, as well as anti-inflammatory (*Mrc1* and *Clec10a*) and oxidative phosphorylation process (*Cox6a2* and *Actn3*) (Fig. 3c), suggesting the typical AIM-driven fibrous tendency in the Ctrl_LW group. In contrast, the ECM_LW sample supports a metabolic profile associated with the glycolysis process (*Eno1*, *Gapdh*). Besides, ECM_LW group expression included gene sets related to the regulation of T cell activation and alpha-beta T cell proliferation that suggested an adaptive immune predominance state in a biomaterial-driven microenvironment (Fig. 3b, c).

To trace the spatial heterogeneity of defined single-cell subclusters, we integrated the expression profiles of sc-RNA seq and ST using AddmoduleScore fuction[32]. The top 50 (based on avg_log2FC) marker genes of cutaneous cells in the single-cell dataset (defined in Fig. 2) were scored and projected into the ST slices. The spatial feature plot illustrated keratinocytes crawling around the implanted biomaterials (Fig. 3d), which was verified in Supplementary Fig. 2a. Intriguingly, fibroblasts from different lineages showed a distinct spatial distribution pattern in the Ctrl_LW group. Upper lineage PF[1] (HF-related) was limited in the upper layer of granulation tissue, while lower lineage RF[1] (fibrous response-related) predominated in the lower layer (Fig. 3e). In contrast, the distribution of PF[1] (HF-related) was enlarged around biomaterials in the ECM_LW group (Fig. 3e), and ECM_LW showed the higher expression level of PF[1]

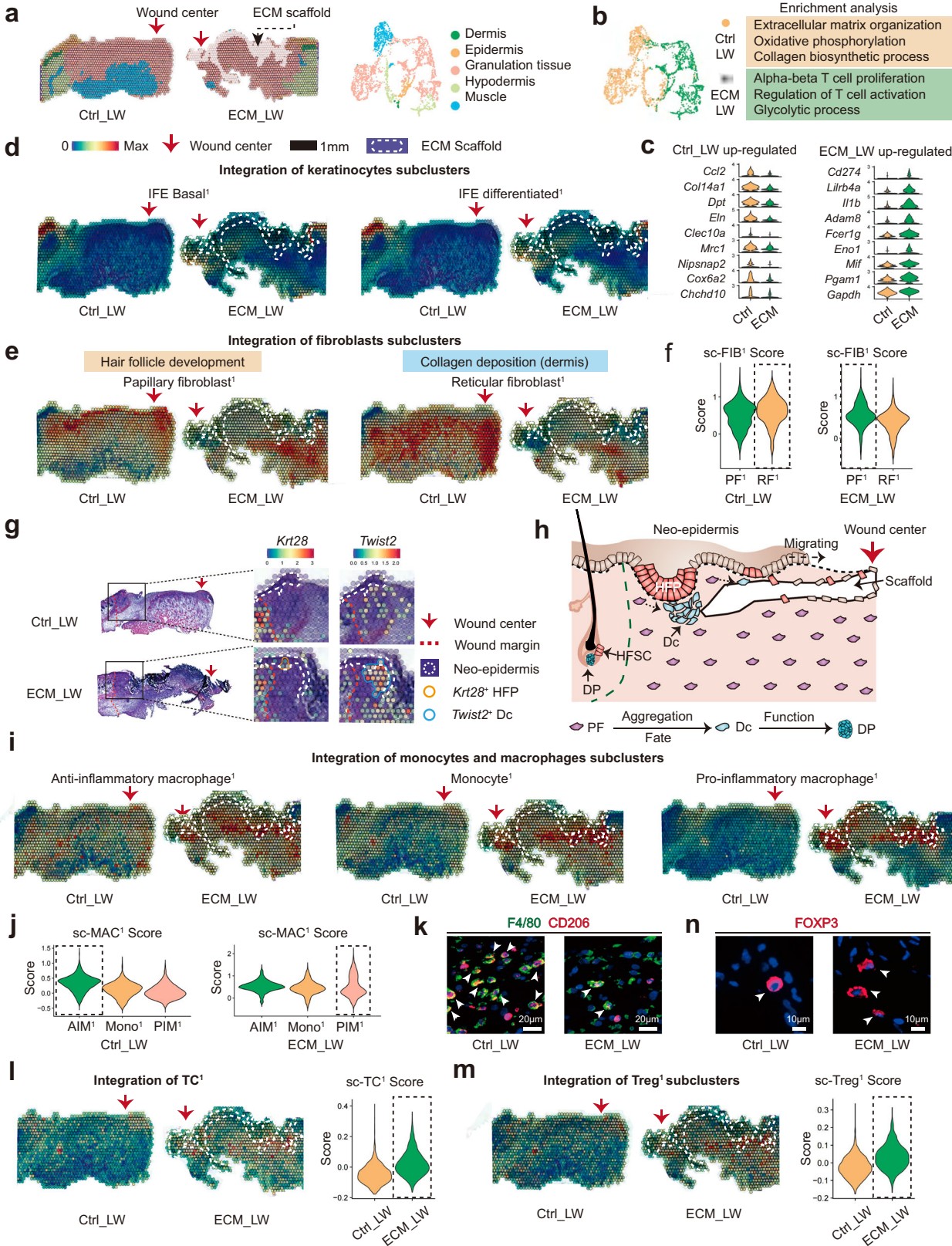

(Fig. 3f), suggesting the pro-regenerative potential of ECM sample. In addition, in the ST profiling of the ECM_LW group, we identified the initial structure of the hair germ (Fig. 3g) composed of HFP in the neo-epidermis and Dc aggregation beneath the dermis (verified by IF in Fig. 1g), which highlighted the early HF formation signature in the ECM_LW group at the spatial gene expression level (Fig. 3h).

To assess the spatial characteristics of scaffold-induced immune microenvironment, we integrated the immune cells subclusters (defined in Fig. 2) and ST slices in like manner. Regarding MAC subclusters, ECM showed the ability to reduce the recruitment of AIM[1] (Fig. 3i, j). IF staining of F4/80 and CD206 confirmed the reduced AIM[1] in the ECM_LW group (Fig. 3k), which might relieve the subsequent

**Fig. 3 | Spatial anchors tracing the cell distribution around the ECM scaffold.**
**a** The unsupervised clustering indicated the anatomical structure of samples.
**b** Gene enrichment analysis between ECM_LW and Ctrl_LW group. **c** Violin plot
showing the up-regulated genes of Ctrl_LW and ECM_LW samples of ST profile.
**d** Spatial feature plot showing the distribution of IFEB[1] and IFED[1] subclusters in
tissue sections. **e** Spatial feature plot showing the distribution of PF[1] and RF[1] sub-
clusters in tissue sections. **f** Violin plots of FIB scores of individual spots derived
from scRNA-seq data (sc-FIB score) for each subcluster. Dotted boxes stressed
clusters with the higher average sc-FIB scores. **g** The spatial feature plot highlighted
the expression of *Krt28*[+] hair follicle progenitor and *Twist2*[+] dermal condensate in
migrating neo-epidermis. **h** Illustration showing the epithelialization along with de

novo HF formation in the biomaterials-mediated healing process. **i** Spatial feature
plot showing the distribution of MAC subclusters in tissue sections. **j** Violin plots of
MAC scores of individual spots derived from scRNA-seq data (sc-MAC score) for
each subcluster. Dotted boxes stressed clusters with the highest average sc-MAC
score. **k** Representative IF images of stained AIM[1] (F4/80[+]CD206[+]), white arrow-
heads showing the F4/80[+]CD206[+] cells. **l** Spatial feature plot and violin plot showing
TC[1] distribution and expression level in tissue sections. **m** Spatial feature plot and
violin plot showing the distribution and expression level of Treg[1] in tissue sections.
**n** Representative IF images of stained Treg[1] (FOXP3), white arrowheads showing the
FOXP3[+] cells.

fibroblast activation and collagen deposition. Instead, we noticed the
apparent aggregation of T cells (Fig. 3l), especially Treg[1] (Fig. 3m),
colocalized with PF[1] (HF-related) surrounding the biomaterial. The
recruitment of Treg[1] (confirmed by IF in Fig. 3n) might contribute to
the suppression of the type-2 immunity activated by AIM[1].

**Prediction of critical signaling patterns between spatially co-
localized cell populations around the biomaterial**
Based on the multimodal profiling, we observed a higher proportion of
T cells (the schematic summary was shown in Fig. 4a) and HF precursor
cells in the ECM_LW group, which implied a potential cellular com-
munication among them. Next, we applied CellChat to predict the cell-
cell communication patterns of immune cells and cutaneous cells
between ECM_LW and Ctrl_LW groups. The circle plots showed the
overall interaction number of the ligand-receptors in the Ctrl_LW and
ECM_LW groups (Fig. 4b). An intensive communication network
between fibroblasts and immune cells was observed in both groups.
The more abundant interactions in the Ctrl_LW group appeared to be
sent from the RF[1], LF[1], MF[1], and Macf[1] subpopulations, indicating the
predominance of fibrotic fibroblasts interaction signals in the dermis.
In contrast, the more abundant interactions in the ECM_LW group were
sent from the Dc[1] and PF[1] subpopulations, indicating the pro-
regenerative signals induced by biomaterials. Since AIM[1] might be
the major immune cells contributing to tissue fibrosis, we compared
specific interactions among AIM[1] and fibroblast subpopulations
(Fig. 4c). The interaction of AIM[1] with fibrotic fibroblasts was more
noted in the Ctrl_LW group via *Tgfb1–(Tgfbr2+Acvr1b)* binding, which
play potential pro-fibrotic roles on the target cells[13]. In comparison, the
interaction of Treg[1] with HFSC was more noted in the ECM_LW group
via *Jag1–Notch1* binding, suggesting the pro-regenerative role of Treg[1]
in biomaterial-treated wounds[60]. Of note, we detected the more sig-
nificant interactions of Treg[1] with PIM[1] and Mono[1] via *Jag1–Notch2*
binding. It has been reported the essential role of Notch signaling in
macrophage polarization. Selective inhibitors of Notch signaling sig-
nificantly suppressed M1-like macrophages and up-regulated the M2-
like macrophages[62]. Consistent with the gene expression profile at the
single-cell level, we confirmed the potential *Jag1–Notch2* communica-
tion in ST profiles (Fig. 4e). Prominent *Notch2* expression was observed
around implanted biomaterial, implicating the suppressive role of
Tregs in the scaffold-mediated microenvironment.

**The adaptive immune system was required for the skin regen-
eration mediated by ECM scaffolds**
To determine the role of the adaptive immune system in HF regenera-
tion, we placed ECM scaffolds in the dorsal skin of immunodeficient
C57Bl/6 (B6.129-Rag2tm1) mice, which lacked mature T lymphocytes[63,64]
(Fig. 5a). We noticed that the wound closure rate was delayed on POD7
(Fig. 5b–d), and regeneration of HF was scarce (Fig. 5b, e) on day 28 in
Rag2[-/-] mice. Compared to wildtype (WT) mice, Rag2[-/-] group still
possessed a myeloid recruitment ability, but no visible accumulation of
T cells around biomaterials was observed (Supplementary Fig. 7). Next,
we further explored the differences in cell composition and spatial gene
expression between WT and Rag2[-/-] mice using scRNA-seq

(Supplementary Fig. 8). As shown in Fig. 5f, in accordance with the HF
regeneration outcomes, the Rag2[-/-] group reduce the proportion of PF[2]
(related to HF development) but improve the number of RF[2] (related to
collagen deposition) and LF[2] (related to collagen deposition and
angiogenesis). For the immune microenvironment, the Rag2[-/-] samples
recruited fewer T cells but more monocyte/macrophages than the WT
group (Supplementary Fig. 8b). T cell related genes such as *Areg*, *Trdc*,
and *Rgs2* were down-regulated in Rag2[-/-] samples (Supplementary
Fig. 8c). The proportion of neutrophil and dendritic cell subsets was
basically equilibrium between the two groups (Supplementary Fig. 9).
Subclustering of MAC resulted in three subsets including PIM[2], AIM[2],
and Mono[2] (Fig. 5g). Rag2[-/-] samples improved the recruitment of AIM[2],
PIM[2], in which AIM[2] owned the highest proportion (46.2%). Sub-
clustering of T cells resulted in seven subsets (Fig. 5h). The Rag2[-/-]
groups contributed to fewer T cells in most subsets except NKT[2]
(related to natural killer cell-mediated cytotoxicity) and naive Th2[2],
indicating the dysfunctional adaptive immune systems in Rag2[-/-] group.
The absence of suppressive Treg[2] might be the reason for uncontrolled
type-2 AIM[2] accumulation and collagen deposition of fibrotic RF[2].

To explore the spatial characteristics of cell heterogeneity in
immunodeficient mice, we also applied ST to analyze the spatial gene
expression profiles. Anatomical structures could be identified in the ST
sections (Fig. 6a). Based on the gene enrichment analysis of ST, the
Rag2[-/-] sample down-regulated gene expression in hair follicle mor-
phogenesis but improved the genes related to collagen fibril organi-
zation (Fig. 6b). Compared to WT samples, the distribution and
expression level of PF[2] (HF-related) was reduced (Fig. 6c), which was
verified by the IF staining of the corresponding marker gene *Crabp1*
(Fig. 6d). In contrast, the distribution and expression level of lower
lineage *Mest*[+] RF[2] were significantly increased (Fig. 6e, f), in accordance
with the healing outcome of Rag2[-/-] groups. To compare the scaffold-
induced immune microenvironment, we also integrated marker genes
of immune cells (defined in Fig. 5) with ST profiling. We confirmed the
more obvious aggregation of AIM[2] surrounding the biomaterial in
the Rag2[-/-] sample (Fig. 6g, h), which might contribute to the absence of
the Treg[2] (Fig. 6i, j).

**Biomaterials facilitate de novo HF regeneration despite
wound size**
To detect any differences in wound healing between small and large
wounds treated with biomaterials, we repeated the experiment with
small (diameter=0.6 cm) full-thickness wounds implanted with bio-
materials (Fig. 7a, b). Consistent with the ECM_LW group, the scaffold
implanted in the small wound (ECM_SW) did not trigger an obvious
FBR fibrous capsule and had a rapid degradation rate (Fig. 7c). Histo-
logical sections of ECM_SW tissue revealed clear signs of enhanced HF
reconstruction too (Fig. 7c, d). ST was also applied to explore the
spatial characteristics of Ctrl_SW and ECM_SW samples, and the ana-
tomical structure was shown in Fig. 7e. We still observed the recruit-
ment of *Cd3*[+] T cells around the implanted biomaterial, which were co-
located with *Crabp1*[+] papillary fibroblasts in ECM_SW sample (Fig. 7g),
confirming the pro-regenerative potential of ECM scaffolds despite of
wound size.

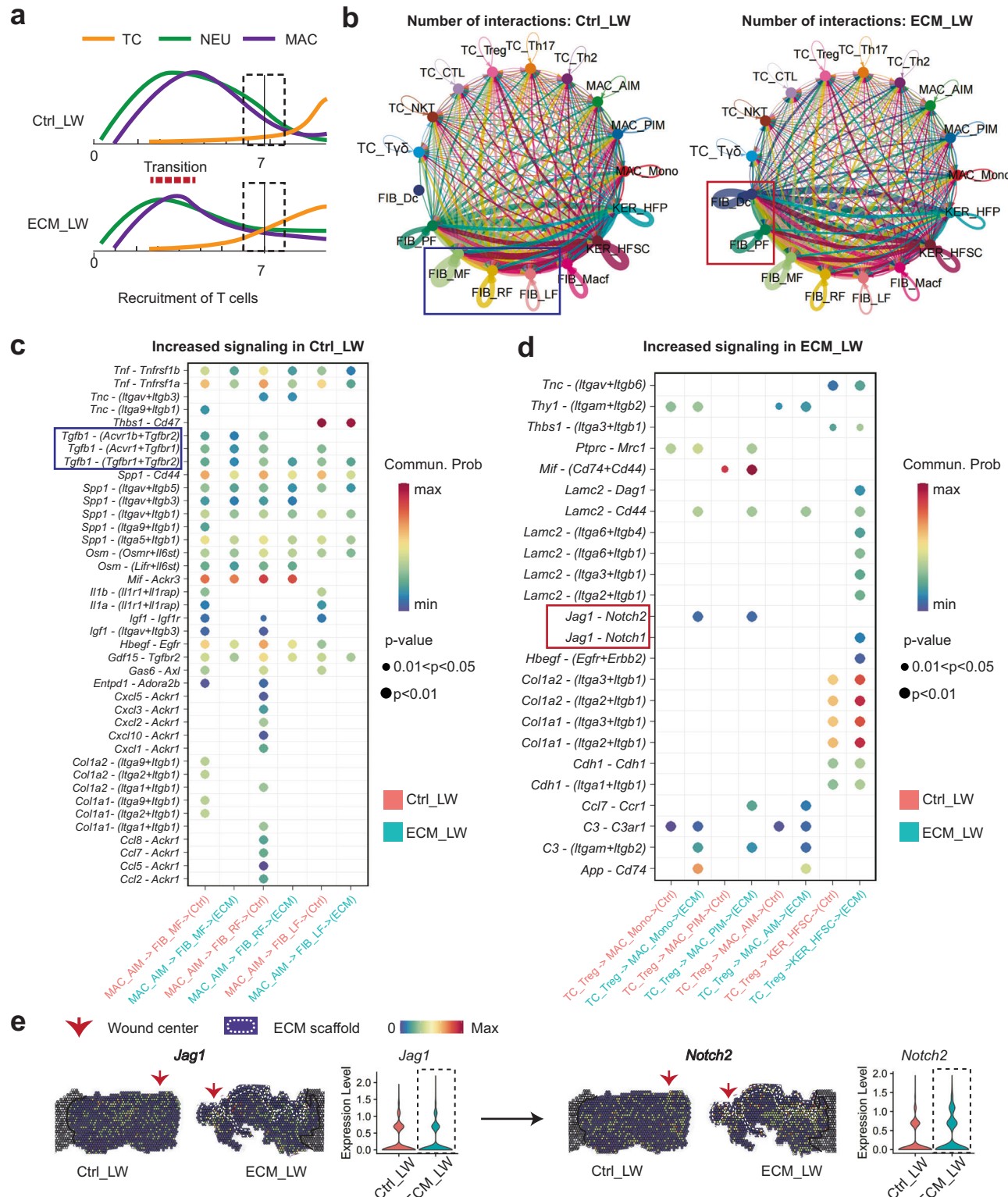

**Fig. 4 | Cellular communication landscape between immune cells and cutaneous cells. a** Schematic timeline highlighting the recruitment of immune cells from innate and adaptive immune systems in Ctrl_LW (top) and ECM_LW (bottom) groups. **b** Comparison of overall cell-cell interaction numbers of immune cells and cutaneous cells between Ctrl_LW and ECM_LW using CellChat. **c** The

ligand–receptor pairs up-regulated in the Ctrl_LW group in specificity between AIM[1] and fibroblasts (MF[1], RF[1], and LF[1]). **d** The ligand–receptor pairs up-regulated in the ECM_LW group in specificity between Treg[1], MAC (Mono[1], PIM[1], AIM[1]), and HFSC[1]. **e** Spatial feature plots and corresponding violin plots showed the expression level of the ligand and cognate receptor in the Notch signaling pathway.

## Discussion

Synthetic graft transplantation is an efficient treatment for severe large-area skin wounds, especially when the donor site is not qualified[18]. With an additional understanding of tissue engineering, it is

suggested that synthetic biomaterials imitating the native structure of ECM can integrate and potentially play a pro-regenerative role in wound healing[20,65]. The reconstruction of dermal appendages is an essential indicator of complete skin regeneration. Nevertheless,

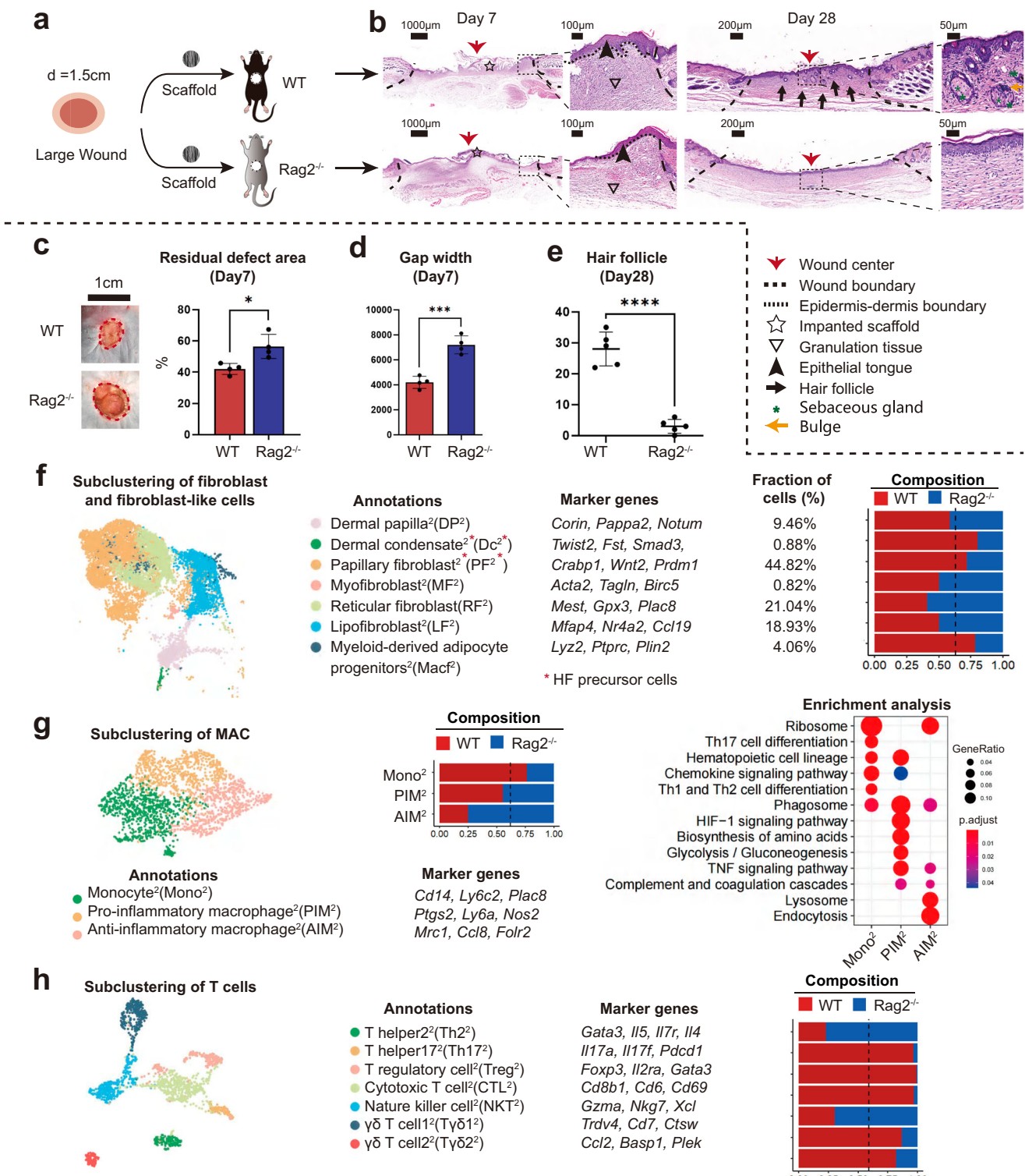

**Fig. 5 | Evaluation of wound healing in immunodeficient mice lacking mature T cells. a** The surgical process for evaluating large area wound healing mediated by ECM scaffold in WT and Rag2⁻/⁻ mice. **b** Representative histological images of wound healing in WT and Rag2⁻/⁻ mice at 7 and 28 days. **c** Residual defect area on POD 7 (Data are presented as mean ± SD, $n = 4$ biologically independent samples, two-tailed t-test, *$p = 0.014$). **d** Semiquantitative evaluation of gap width (Data are presented as mean ± SD, $n = 4$ biologically independent samples, two-tailed t-test, ***$p = 0.00046$). **e** Histologic quantification of de novo HFs (Data are presented as mean ± SD, $n = 5$ biologically independent samples, two-tailed t-test, ****$p = 0.000013$). **f** Subclustering of fibroblasts and fibroblast-like cells showing two fibroblast-like subsets and five fibroblast subsets. Marker genes for fibroblast subsets are listed. **g** Subclustering of monocyte/macrophage showing three subsets. The marker genes, composition, and enrichment analysis for each subset are listed. **h** Subclustering of T cells showing seven subsets. The marker genes and composition for each subset are listed. $p$ value: *$p < 0.05$, **$p < 0.01$, ***$p < 0.001$ and ****$p < 0.0001$.

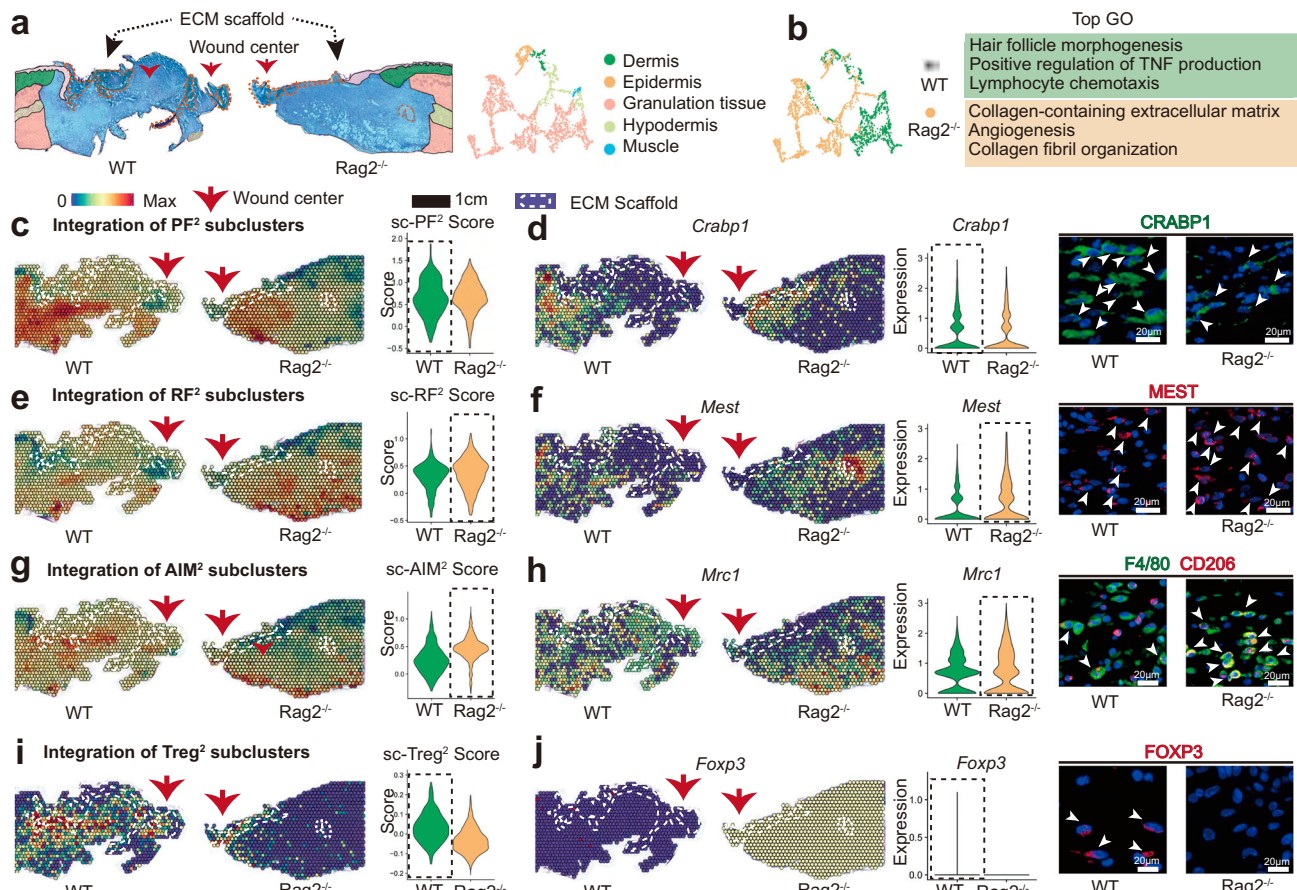

**Fig. 6 | Spatial atlas of cell microenvironment around biomaterials of immunodeficient mice. a** The anatomical structure of each sample. **b** Gene enrichment analysis between WT and Rag2[-/-] group. **c** Spatial feature plot and violin plot showing the distribution and expression level of integrated PF[2] subcluster in tissue sections. **d** Spatial feature plot and violin plot showing the distribution and expression level of *Crabp1* (marker gene of PF[2]) in tissue sections; Representative IF images of stained PF[2] (CRABP1[+]), white arrowheads showing the CRABP1[+] cells. **e** Spatial feature plot and violin plot showing the distribution and expression level of integrated RF[2] subcluster in tissue sections. **f** Spatial feature plot and violin plot showing the distribution and expression level of *Mest* (marker gene of RF[2]) in tissue sections; Representative IF images of stained RF[2] (MEST[+]), white arrowheads showing the MEST[+] cells. **g** Spatial feature plot and violin plot showing the distribution and expression level of integrated AIM[2] subcluster in tissue sections. **h** Spatial feature plot and violin plot showing the distribution and expression level of *Mrc1* (marker gene of AIM[2]) in tissue sections; Representative IF images of stained AIM[2](F4/80 [+]CD206[+]), white arrowheads showing the F4/80 [+]CD206[+] cells. **i** Spatial feature plot and violin plot showing the distribution and expression level of integrated Treg[2] subcluster in tissue sections. **j** Spatial feature plot and violin plot showing the distribution and expression level of *Foxp3* (marker gene of Treg[2]) in tissue sections; Representative IF images of stained Treg[2] (FOXP3[+]), white arrowheads showing the FOXP3[+] cells.

existing treatments can only form epidermal or dermal layers and fail to regenerate enough HF and SG[66]. Requirements for the promotion of nascent HF formation bring significant challenges to biomaterial design, particularly for large-scale severe wounds.

Here, the pro-regenerative influence of the adaptive immune system in coordinating wound repair has been stressed[12]. We employed an ECM scaffold for large-area skin defects and investigated its immunoregulatory mechanism in wound healing. The scaffold showed an impact by accelerating wound closure and promoting nascent HF formation (Fig. 1c–g). By multimodal analysis, we observed the substantial accumulation of type 2 immune cells (especially *Mrc1*[+] AIM and *Gata3*[+] Th2 cells) in Ctrl_LW samples (Fig. 2d, e), which might promote the fast healing of damaged tissue at the expense of the original skin composition and function[67,68]. In contrast, adaptive T cell infiltration in response to scaffold implantation was driven towards immunosuppression subpopulations in the ECM_LW group. A larger number of *Foxp3*[+] Tregs were recruited by ECM scaffold to mitigate skin fibrosis by suppressing excessive type 2 macrophage inflammation (Figs. 2e and 3l–m). We next confirmed the requirement of T cells in skin regeneration by an immunodeficient model (Fig. 5). The absence of suppressive Treg[2] might be one of the reasons for the uncontrolled accumulation of type-2 AIM[2], which might drive collagen deposition by activating profibrotic RF[2]. These data validated that the activation of adaptive immune was required for the reparative effect mediated by ECM scaffolds. Meanwhile, mild FBR around the scaffold decreased the risk of immune rejection, and the appropriate degradation rate could also avoid the extra expense of secondary surgery in both small and large-area wound healing (Fig. 7).

In this study, we offered an available manner for large-area wound regeneration and first defined the spatial heterogeneity of the microenvironment during the biomaterial-mediated wound healing process. These techniques provided a unique medium through which we can further understand the immunoregulatory mechanisms of the ECM scaffold. Of note, we testified the role of the adaptive immune system activated by biomaterials in HF reconstruction and provided further insights into the future design of targeted immunoregulatory materials for scarless wound regeneration.

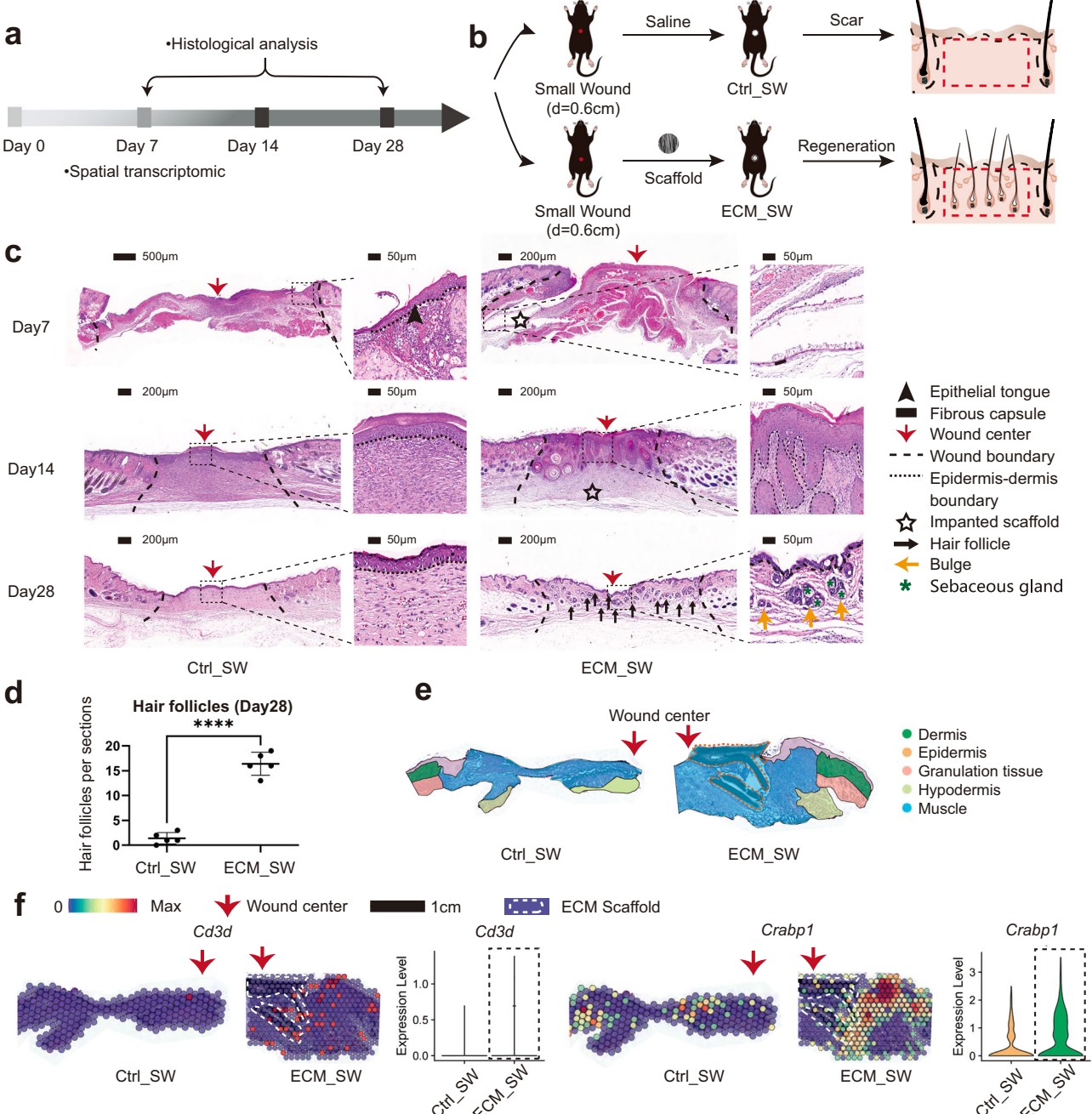

**Fig. 7 | Evaluation of the healing of small full-thickness wounds treated with ECM scaffolds. a** Workflow for evaluating skin wound healing. **b** The surgical process for skin excisional wound model of Ctrl_SW and ECM_SW group. **c** Representative H&E images of Ctrl_SW and ECM_SW samples. **d** Histologic quantification of de novo HFs on POD28 (Data are presented as mean ± SD, $n = 5$ biologically independent samples, two-tailed t-test, ****$p = 0.000001$). **e** The anatomical structure of samples. **f** Spatial feature plot showing the expression of *Cd3d* (marker gene of T cells) and *Crabp1* (marker gene of papillary fibroblasts) in ST profile and corresponding quantitative analysis. *p* value: *$p < 0.05$, **$p < 0.01$, ***$p < 0.001$, and ****$p < 0.0001$.

## Methods

### Ethical approval
All procedures were approved by the Institution Review Board of West China Hospital of Stomatology (No. WCHSIRB-D-2020-385).

### Fabrication of ECM scaffolds
To fabricate the electrospinning scaffold with approximate 300 nm diameter aligned fibers, 20% w/v PLGA (LA/GA = 75:25, molecular weight = 105 kDa, Jinan Daigang Biomaterial Co. Ltd.) and 2% w/v FC (Sangon Biotech Co. Ltd.) were dissolved in HFIP (Aladdin Co., Ltd.) with stirring until complete dissolution. The solution was fed at

0.018 mm/min, and a voltage of −2/5kv and 11 cm distance between the needle (21 G) and the rotating cylinder (2800 rpm) was applied. After been dried in a vacuum oven for a week. The morphology of PLGA/FC nanofibrous scaffold was observed by scanning electron microscopy (SEM; JSM-7500F, JEOL, Japan). The scaffolds were cut into circular shapes (0.8 cm or 1.8 cm in diameter) and sterilized using γ-irradiation before the animal implantation experiments.

### Excisional wound model and implantation procedures
All procedures involving animals were approved by the Institution Review Board of West China Hospital of Stomatology

(No. WCHSIRB-D-2020-385). Female wildtype C57BL/6 J mice (Dossy Experimental Animals Co., Ltd.) and immunodeficient C57Bl/6 (B6.129-Rag2tm1) mice (Shanghai Model Organisms Center Inc., Shanghai, China), at the age of 6–8 week ( ~ 20 g) were used in this research. The mice were housed under standard conditions including temperature of 21–27 °C, humidity of 40–70%, and a 12 h light-dark cycle with free access to food. The number of animals used for each experiment is indicated in the figure legends. To minimize wound contraction by the panniculus carnosus of rodents and allow wound healing through granulation and re-epithelialization like the human skin, we use the mice-splinted model[36]. The circular (diameter =0.6 or 1.5 cm) full-thickness wounds were created in the mice dorsal skin and stented by silicone loops. The mice in further study were divided into three groups: large wound (diameter = 1.5 cm) treated with saline (Ctrl_LW) or ECM scaffolds below the wound (ECM_LW), and small wound (diameter = 0.6 cm) treated with ECM scaffolds (ECM_SW). Subsequently, the wounds were covered with sterile Tegaderm film (3M) and respectively fixed on the small (inner diameter 8 mm and outer diameter 12 mm, for small wound) or large (inner diameter 18 mm and outer diameter 22 mm, for large wound) silicone ring with uniform suture. Mice were euthanatized at 1–4 weeks after the surgery, and the small (diameter 10 mm, for small wound) or large (diameter 25 mm, for large wound) round full-thickness sample was harvested.

## Bulk-RNA sequencing

Three replicates of mice skin wounds in each group were collected for the of bulk-tissue RNA sequencing procedure. Total amounts and integrity of RNA were assessed using the RNA Nano 6000 Assay Kit of the Bioanalyzer 2100 system (Agilent Technologies, CA, USA). mRNA was purified from total RNA by using poly-T oligo-attached magnetic beads. The library fragments were purified with AMPure XP system (Beckman Coulter, Beverly, USA). The PCR amplification product was purified by AMPure XP beads. After the library is qualified by qRT-PCR, the different libraries are pooling and being sequenced by the Illumina NovaSeq 6000. FeatureCounts (v1.5.0-p3) was used to count the reads numbers mapped to each gene. And then FPKM of each gene was calculated based on the length of the gene and reads count mapped to this gene. Differential expression analysis of two groups (three biological replicates per group) was performed using the DESeq2 R package (v1.32.0), $p$ value< 0.05, and |$\log_2$(foldchange)| >2 was set as the threshold for significantly differential expression. Gene enrichment analysis was performed using ClusterProfiler (v4.6.2) and org.Mm.eg.db (v3.13.0). GO terms and KEGG pathways with corrected $p$ value less than 0.05 were considered significantly enriched by differentially expressed genes. Chord plots and heatmap of differentially expressed genes were plotted by GOplot (v1.0.2) and ggplot2 (v3.4.2).

## Single cell RNA sequencing

**Tissue dissociation.** Three fresh samples were collected per group for scRNA-seq. In brief, the wound tissues were firstly digested by the Epidermis Dissociation Kit (Epidermis Dissociation Kit, mouse; Miltenyi Biotec) for enzymatic epidermal-dermal separation. The epidermis part was dissociated by a gentleMACS Dissociator (Miltenyi), then filtered (70-mm cell strainer, Corning, Durham), centrifuged (300 g, 10 min, 4 °C), and resuspended with phosphate-buffered saline (PBS) containing 0.5% bovine serum albumin (BSA). The dermis part was cut into 1 mm width pieces and mixed with mixed enzyme solution containing type I collagenase (Gibco, Grand Island) and trypsin (Gibco, Canada), then dissociated by gentleMACS Dissociator (Miltenyi), and digested for 2.5 hours in a hybridization oven (Peqlab PerfectBlot). After being dissociated, filtered, centrifuged, and resuspended in red blood cell lysis buffer (Solarbio), the dermis cells were mixed with the epidermis part. Then the dead cells and debris were removed by Dead Cell Removal MicroBeads (Miltenyi).

**Sequencing and data processing.** Single-cell suspensions were then carried out for Single-Cell RNA-seq (10x Genomics Chromium Single Cell Kit). Sequencing (10x Genomics Chromium Single Cell 3′ v3) was performed using an Illumina 1.9 mode. Then, reads were aligned, and expression matrices were generated for downstream analysis (Cell Ranger pipeline software).

**Downstream computational analysis.** Different samples were merged into one Seurat object using the RunHarmony function of harmony R packages (v0.1.1) to correct the potential batch effect. Filtering, normalization, scaling, and canonical correlation analysis were performed with the Seurat R package (v4.1.0). RunPCA function was used to process the most variable genes determined by the FindVariableGenes function (selection.method = \"vst\", nfeatures = 4000). ElbowPlot function was performed to determine the number of principle components input. RunUMAP function (Seurat) with the first 20 principal components as input was performed for dimensionality reduction. Unsupervised clustering was performed using the FindClusters function of Seurat and clustree R package (v0.5.0) and differentially expressed genes were determined by the FindAllMarkers function for cluster annotation. ClusterProfiler (v4.6.2) was used to perform the gene set enrichment analysis. CellChat (v1.5.0) was used to predict receptor-ligand probability among cell subpopulations.

## Spatial transcriptomics

**Slide preparation.** Fresh samples were rapidly harvested and frozen in OCT. Wound tissues were cryosectioned at −20 degrees onto gene expression slides. Tissue sections on the Capture Areas of the Visium Spatial Gene Expression are fixed using methanol, H&E staining images will be used downstream to map the gene expression patterns. Using the Visium Tissue Optimization Slide & Reagent kit, permeabilization time was optimized. Second Strand Mix is added to the tissue sections to initiate second strand synthesis. After the transfer of cDNA from the slide, spatially barcoded, full-length cDNA is amplified via PCR to generate sufficient mass for library construction. The final libraries contain the P5 and P7 primers used in Illumina amplification. A Visium Spatial Gene Expression library comprises standard Illumina paired-end constructs that begin and end with P5 and P7.Raw FASTQ files and histology images were processed (SpaceRanger software) for genome alignment.

**Downstream computational analysis.** Raw output files for each group were read into R studio with the Seurat R package (v4.1.0). Normalization across spots was performed with the SCTransform function. The spatial cluster gene signature overlap correlation matrix was generated by first taking all genes differentially expressed (avg_log2FC > 1 and adjusted $p$ value < 0.05) across all ST clusters.

**Integration analysis of scRNA-seq and ST.** Signature scoring derived from scRNA-seq and ST signatures was performed with the AddModuleScore function in Seurat R packages (v4.1.2)[32]. Firstly, the FindAllMarkers function was used to identify the top 50 marker genes (based on avg_log2FC value) of each cluster on the single-cell level. AddModuleScore was then used to calculate the average expression of each gene set in each spot of the spatial transcriptome. Finally, the scores were mapped to the spatial transcriptome using the SpatialFeaturePlot function, and quantitative assessment was performed using the ggviolin function in ggplot2 R packages (v3.4.2).

**R analysis packages.** R v4.1.0 was used for downstream analysis of single-cell RNA sequencing and spatial transcriptomics data. R packages used: Seurat(v4.1.0), harmony (v0.1.1), clustree (v0.5.0), tidyverse (v2.0.0), Matrix (v2.0.0), ggplot2 (v3.4.2), DEseq2 (v1.32.0), GOplot (v1.0.2), reshape2 (v1.4.4), stringr (v1.5.0), EnhancedVolcano (v1.10.0),

ClusterProfiler (v4.6.2), org.Mm.eg.db (v3.13.0), Cellchat (v1.5.0), patchwork (v1.1.2), data.table (v1.14.8), hdf5r (v1.3.8), pracma (v2.4.2). The tutorial of Seurat software is available at https://satijalab.org/seurat/index.html. The tutorial of ClusterProfiler software is available at https://github.com/YuLab-SMU/clusterProfiler. The tutorial of GOplot software is available at https://github.com/cran/GOplot. The tutorial for EnhancedVolcano software is https://github.com/kevinblighe/EnhancedVolcano. The tutorial on Cellchat software is available at https://github.com/sqjin/CellChat.

**Histopathology, Immunohistochemistry, and immunofluorescence microscopy.** The samples were fixed with 4% paraformaldehyde at least 48 hours before ethanol and xylene dehydration. H&E staining and Masson's trichrome staining were performed for the observation of re-epithelialization and collagen fiber deposition. Immunofluorescence staining for cytokeratin 5 (ab52635, Abcam, 1:200), cytokeratin 10 (ab76318, Abcam, 1:150), cytokeratin 17 (17516-1-AP, Proteintech, 1:200), TWIST2 (66544-1-Ig, Proteintech, 1:100), Ki67 (Servicebio, GB121141, 1:100), SCD1 (28678-1-AP, Proteintech, 1:200), CRABP1 (13163 S, Cell Signaling, 1:100), MEST (11118-1-AP, Proteintech, 1:100) were performed for to assess wound regeneration. For the evaluation the infiltration of immune cells, immunohistochemistry staining for CD3 (14-0032-82, Thermo Fisher Scientific, 1:100), CD68 (ab283654, Abcam, 1:100), Ly6G (ab238132, Abcam, 1:2000) and immunofluorescent staining for F4/80 (29414-1-AP, Proteintech, 1:200), CD206 (360017, Zenbio, 1:100), FOXP3 (sc-53876, Santa Cruz Biotechnology, 1:100), were performed.

**Flow cytometry analysis.** Single cells digested from skin wounds were pre-incubated with purified anti-CD16/CD32 antibody (101301, BioLegend) (1.0 μg per $10^6$ cells in 100 μl volume) for 5 to 10 min to block Fc receptors. The cell suspensions were then co-incubated with fixable viability dye (eFluor™ 780, 65-0865-14, eBioscience) and antibodies against surface markers CD45 (PE/Cyanine7, 147703, BioLegend), CD3 (PE, 100205, BioLegend), and F4/80 (FITC, 123107, BioLegend) at 1:400 dilution for 30 min at 4 °C in the dark (100 μl per antibody per sample). After fixation and permeabilization, cells were incubated with antibodies against intracellular marker CD68 (APC, 137007, Biolegend) at 1:400 dilution for 20 min at 4 °C in the dark (100 μl per antibody per sample). Isotype controls of CD45 (PE/Cyanine7 Rat IgG2b, κ, 400617, Biolegend), CD3 (PE Rat IgG2b, κ, 400607, Biolegend), F4/80 (FITC Rat IgG2a, κ, 400505, BioLegend) and CD68 (APC Rat IgG2a, κ, 400511, Biolegend) were used in same concentration. Flow cytometry analysis was performed using Attune Nxt flow cytometer (Thermo Fisher Scientific) and FlowJo (v10.8.1). The experiments were performed three times independently ($n = 3$).

**Statistics and reproducibility.** The precise sample number for each experiment was indicated in the figure legends. For Figs. 1g, 3k, 3n, 6d, 6f, 6h, and 6j, representative images were shown from one of three biological repeats; For Fig. 5b and Supplementary Fig. 2a, representative images were shown from one of four biological repeats; For Figs. 1e, 7c and Supplementary Fig. 1c, 2d, 7a, representative images were shown from one of five biological repeats. The data was dealt with Case Viewer software, Image J software, and Prism 8.0 software. Statistical analyses were performed by two-tailed t-test/ t′-test or one-way analysis of variance (ANOVA) with Tukey post-hoc test. The numerical data were presented as mean ± standard deviation. A value of $p < 0.05$ was considered statistically significant (*$p < 0.05$, **$p < 0.01$, ***$p < 0.001$, ****$p < 0.0001$), and ns means no statistically significant difference.

**Reporting summary**
Further information on research design is available in the Nature Portfolio Reporting Summary linked to this article.

## Data availability
The RNA sequencing data used in this study are available in the NCBI Sequence Read Archive (SRA) database under accession code: "PRJNA1004430". All other data supporting the findings of this study are available within the article and its supplementary files. Any additional requests for information can be directed to, and will be fulfilled by, the corresponding authors. Source data are provided in this paper. Source data are provided with this paper.

## Code availability
Analytical tools, variables, and sources of tutorials were described in the Methods section. R code is available from the corresponding author upon request.

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

## Acknowledgements

The study was funded in part by the National Natural Science Foundation of China, grant 82271015 (Y.M.); Sichuan Science and Technology Program, grant 2022YFS0041 (Y.Q.) & 2022ZYD0052 (Y.M.); Interdisciplinary Innovation Project, West China Hospital of Stomatology, Sichuan University, grant RD-03-202006 (Y.M.). We thank the Novogene company for RNA sequencing work and Yukai Wang for the assistance of bioinformatics analysis, the National Clinical Research Center for Oral Diseases & State Key Laboratory of Oral Diseases for animal experiments, and the Analytical & Testing Center of Sichuan University for the fabrication of biomaterials. We thank Xinhui Li, Jiayu Gao for the help with analysis.

## Author contributions

Y.Y., C.C., Y.M. and Y.Q. conceived the project. Y.Y. designed and performed all the experiments with the assistance of C.C., L.L., C.W. and C.H. Data were analyzed by Y.Y. and S.R. with the help of C.H. The manuscript was written by Y.Y. with the guidance of C.C. and Y.M. The project was supervised by Y.M. and Y.Q. jointly.

## Competing interests

The authors declare no competing interests.
