## [Peer Review File · Nature Communications]

REVIEWER COMMENTS

Reviewer #1 (Remarks to the Author):

In the manuscript 'Tracing immune cells around biomaterials with spatial anchors during large-scale wound regeneration', Yang et al. present a method for wound regeneration using biodegradable ECM scaffold. They provide multimodal analysis during wound regeneration, including scRNA-seq and spatially-informed transcriptomics, to study cellular response around such biomaterials.

In general, the paper could be written more clearly, and requires editing for typos, clarity, etc.

In this review I mostly focus on the single-cell analysis aspect of the work.

- line 183 - 'we integrated marker genes of immune cells with ST profiling' - how was this done? this is a crucial point.

- Elaborate on how were the 'fibroblast-like' cells defined.

- Could you reason about the enrichment analysis for differentially expressed genes in the Ctrl_LW relative to the ECM_LW group based on RNA-seq (Fig. 1i)?

- And the same for the enrichment analysis performed using ST (Fig. 3b)

- It's unclear in which analyses you are using only ST data and in which the integrated ST+scRNA-seq data, clearly define it in the text and figures.

- Many of the claims in the ST-related section, referring to the visualizations in Fig. 3, need to be supported quantitatively, such as the claims corresponding to Fig. 3c, 3d. Same for 3g,h - some can be clearly supported visually and some cannot and therefore it's better to add quantitative assessment in a systematic way.

- Line 192 - you report experimental observations, but the figures you point to (Fig. 4a,b) are schematics.

- Lines 194-5 - it sounds from your phrasing that you aim to predict cell-cell communication between the ECM_LW and Ctrl_LW populations (which is not the case based on Fig. 4c).

- Fig. 4c-e should be explained in detail in the caption.

- What can we learn from the ligand-receptor analysis using the ST data (e.g. Fig. 4d,e)? Can you see a spatial correlation between the expression of ligands and receptors? Any other spatially-based conclusions?

- Authors need to be more explicit/elaborate when they use terms related to the analysis of the data (e.g. clustering - which type of clustering? what parameters were used? same for integration)

- How did the authors batch-correct the single-cell data between the different samples (e.g. ctrl and ECM)

- line 228 - in Fig. 5k there are 8, not 7 clusters shown.

- ST results in Fig. 6 - same comment as above, authors need to support their claims quantitatively (about overall changes in expression and where they occurred in the tissue relative to the wound or relative to other landmarks). And the same for Fig. 7.

- Line 253 - the results need to be further elaborated in the text. For example, authors write 'We further explored the differences in immune cell composition using scRNA-seq and ST (Fig. 7 g, h)' without any explanation or further discussion of the results shown in these subplots (and the same comment is relevant for almost all results shown for ST analysis related to Fig. 7).

- In the methods section the authors should dedicate additional subsections to the computational analysis of the data

Minor comments:

- many of the font sizes in the figures are too small to read

- the legend in fig. 1i is unclear
- line 195: typo in ECM

Reviewer #2 (Remarks to the Author):

Skin scarring without dermal appendage formation is a common problem in wound healing. The manuscript deals with this by characterizing the effect of a biodegradable aligned extracellular matrix (ECM) scaffold. They observed faster wound closure and enhanced hair follicle neogenesis by the implantation of ECM scaffold in mouse excisional wounds. This study then utilized single-cell RNA sequencing and spatial transcriptomics to understand the effect of biodegradable ECM scaffold during wound healing with a focus on immune cell populations and hair follicle neogenesis. They also showed that immunodeficient mice (Rag2 $-/-$) show defects in hair follicle neogenesis. However, the project is very descriptive. As scRNA seq of hair follicle neogenesis has been published by several groups, the novelty of this study is limited. More specific and in-depth understanding of how components of ECM bring about cellular and molecular changes would be necessary to establish a solid novel model/theory in this field.

1. Several studies indicate the incidence and extent of hair follicle neogenesis vary between samples. Quantifications for hair follicle regeneration need to be more rigorous. What percentage of the samples contained de novo hair follicle in each condition? Picture of healing time course and hair follicle neogenesis in control without splint needs to be demonstrated.
2. The reasoning for examination of Day 7 post wounding in scRNA seq analyses is insufficient. ECM may change the cell composition at other time points and influence the cell state of wound healing.
3. If omics suggest hair neogenesis occurs on day 7 prior to re-epithelialization with ECM biomaterials, such examples need to be verified by IF or histology.
4. How does the ECM scaffold bring about cellular and molecular changes? Specific mechanisms of how ECM regulates specific immune cells need to be analyzed in association with functional data.
5. The requirement of adaptive immune cells was shown with Rag2 $-/-$ mice. How ECM regulates adaptive immune cells needs to be analyzed in more depth.
6. Previous studies showed the roles of macrophages and gamma delta T cells are essential for hair follicle neogenesis. Previous studies showed peripheral parts of large wounds can induce hair follicle neogenesis with shh signal activation. Relation to these previous studies is unclear.
7. The authors need to provide sufficient introduction regarding hair follicle neogenesis and immune response in skin wounds.

Reviewer #3 (Remarks to the Author):

Yang et al present a thorough and interesting study on spatial and single-cell and spatial transcriptomic profiling of the immune landscape of skin injury after treatment with a biologic (ECM) scaffold in a mouse model. The study is of interest to a general audience within the field of wound healing and adds important state-of-the-art technological advances in characterizing immunomodulation by biomaterials. There are several considerations to be made prior to publication:

Major Comments

- Standard wound healing models are usually 5mm, is there a reason why such a large defect was chosen?
- For the spatial transcriptomics data, often transcriptome deviates from proteome. Immunofluorescence images of a related protein would be useful to place next to the transcriptome visualization. There is some immunohistochemistry and IF but expanding on this would strengthen the conclusions from the spatial transcriptomics.
- Figure 1j – the data suggesting there are more CD3+ T cells present than CD68+ macrophages is quite contradictory to much of published literature. Confirmation with another technique such as flow cytometry would be needed (ex. CD45, CD3, CD68, F4/80)
- Fig 1j – it looks like there is more CD68 staining in the ECM scaffold group but the quantification states otherwise. Is this background staining? Please provide primary delete (secondary only) controls for these stains.

Minor comments

- Line 66 – what does LW stand for? Large wound? Please state in explicitly in text
- Subject-verb agreement and other small grammar items should be double-checked
- Line 73 – what does POD stand for? Please state explicitly in text
- It is hard to evaluate the histology images, please provide high-resolution histology
- Fig 3 – c,d,g,h need labels on Control v ECM scaffold (as in panel a)
- Fig 5 – histopathology images (g) are very small, would suggest moving panel h to supplement (one would expect fewer T cells in a Rag-/- mouse, the confirmation is good for supplemental) and increasing the size of panel g.
- Fig5i – it would not be number (No.) of cells, but Fraction of cells since you are reporting in %

Replies to the Reviewers' Comments

Referee 1

Reviewer #1 (Remarks to the Author):

In the manuscript 'Tracing immune cells around biomaterials with spatial anchors during large-scale wound regeneration', Yang et al. present a method for wound regeneration using biodegradable ECM scaffold. They provide multimodal analysis during wound regeneration, including scRNA-seq and spatially-informed transcriptomics, to study cellular response around such biomaterials.

In general, the paper could be written more clearly, and requires editing for typos, clarity, etc.

Response: We thank Reviewer 1 for reviewing our manuscript and providing insightful and constructive comments so that we can improve it further. We have carefully addressed these concerns and properly revised the manuscript. To enhance clarity, we have supplemented a substantial amount of new data and additional methodological details. In addition, we have checked the manuscript carefully and revised the typos as suggested. Below, we address the questions and suggestions raised by Reviewer 1 point-by-point.

1) In this review I mostly focus on the single-cell analysis aspect of the work. - line 183 - 'we integrated marker genes of immune cells with ST profiling' - how was this done? this is a crucial point.

Response: Thanks for the valuable comment. We have supplemented the methodological details of the integration analysis as suggested. The related descriptions were provided in the Result and Online Methods section of the manuscript with changes marked.

Result (Page 11; Line 244-247): To trace the spatial heterogeneity of defined single-cell subclusters, we integrated the expression profiles of sc-RNA seq and ST using AddmoduleScore function³². The top 50 (based on avg_log2FC) marker genes of cutaneous cells in the single-cell dataset (defined in Fig. 2) were scored and projected into the ST slices.

Online Methods (Page 19; Line 473-480):

Integration analysis of scRNA-seq and ST

Signature scoring derived from scRNA-seq and ST signatures was performed with the AddModuleScore function in Seurat R packages (version 4.1.2)³². Firstly, the FindAllMarkers function was used to identify the top 50 marker genes (based on avg_log2FC value) of each cluster on the single-cell level. AddModuleScore was then used to calculate the average expression of each gene set in each spot of the spatial transcriptome. Finally, the scores were mapped to the spatial transcriptome using the SpatialFeaturePlot function, and quantitative assessment was performed using the ggviolin function in ggplot2 R packages (version 3.4.2).

Reference: [32] Ji, A. L. *et al.* Multimodal Analysis of Composition and Spatial Architecture in Human Squamous Cell Carcinoma. *Cell* **182**, 497-514.e422, doi:10.1016/j.cell.2020.05.039 (2020).

2) *Elaborate on how were the ‘fibroblast-like’ cells defined.*

Response: Thanks for the valuable comment. We have supplemented the detailed elaboration of ‘fibroblast-like’ cells as suggested. The related descriptions were provided in the Result section of the manuscript with changes marked, and the corresponding figure was supplemented in Supplementary Fig. 5a.

Result (Page 7-8; Line 163-172): Besides, skin also contains the specialized

fibroblast-like cell, including dermal papilla (DP) and dermal condensate (Dc), which have unique transcriptional characteristics with universal fibroblast⁴⁶. DP locates at the base of mature hair follicles and serves as the principal signaling niche of hair follicle activities. Origin from PF, dermal condensate (Dc) is believed to be the progenitor of DP in embryonic development^{44,47}. In this dataset, we defined five fibroblasts and two fibroblast-like cells subclusters based on defined markers published before^{22,43,48,49} (Fig. 2c). All subclusters expressed pan-fibroblast marker platelet-derived growth factor receptor- α (Pdgfra)⁴³, while fibroblast-like cells showed the lower expression level of dermatopontin (Dpt)⁵⁰ and higher expression of pappalysin 2 (Pappa2)⁴⁶ (Supplementary Fig. 5a).

Supplementary Figure 5a: Feature plot of the marker genes of pan-fibroblasts (Pdgfra), fibroblasts (Dpt), and fibroblast-like cells (Pappa2).

Reference :

- [22] Hu, C. *et al.* Dissecting the microenvironment around biosynthetic scaffolds in murine skin wound healing. *Science advances* **7**, doi:10.1126/sciadv.abf0787 (2021).
- [43] Driskell, R. R. *et al.* Distinct fibroblast lineages determine dermal architecture in skin development and repair. *Nature* **504**, 277-281, doi:10.1038/nature12783 (2013).

- [44] Mok, K. *et al.* Dermal Condensate Niche Fate Specification Occurs Prior to Formation and Is Placode Progenitor Dependent. *Developmental cell* **48**, 32-48.e35, doi:10.1016/j.devcel.2018.11.034 (2019).
- [46] Joost, S. *et al.* The Molecular Anatomy of Mouse Skin during Hair Growth and Rest. *Cell stem cell* **26**, 441-457.e447, doi:10.1016/j.stem.2020.01.012 (2020).
- [47] Kim, D. *et al.* Noncoding dsRNA induces retinoic acid synthesis to stimulate hair follicle regeneration via TLR3. *Nature communications* **10**, 2811, doi:10.1038/s41467-019-10811-y (2019).
- [48] Abbasi, S. *et al.* Distinct Regulatory Programs Control the Latent Regenerative Potential of Dermal Fibroblasts during Wound Healing. *Cell stem cell* **27**, 396-412.e396, doi:10.1016/j.stem.2020.07.008 (2020).
- [49] Guerrero-Juarez, C. F. *et al.* Single-cell analysis reveals fibroblast heterogeneity and myeloid-derived adipocyte progenitors in murine skin wounds. *Nature communications* **10**, 650, doi:10.1038/s41467-018-08247-x (2019).
- [50] Buechler, M. B. *et al.* Cross-tissue organization of the fibroblast lineage. *Nature* **593**, 575-579, doi:10.1038/s41586-021-03549-5 (2021).

3) Could you reason about the enrichment analysis for differentially expressed genes in the Ctrl_LW relative to the ECM_LW group based on RNA-seq (Fig. 1i)?

Response: Thanks for the valuable comment. We have supplemented the detailed elaboration of enrichment analysis for differentially expressed genes in RNA-seq as suggested. The related descriptions were provided in the Result section of the manuscript with changes marked and the corresponding figure was supplemented in Supplementary Fig. 2 b and c.

Result (Page 5-6; Line 112-130): To explore the underlying mechanism, a bulk-tissue RNA-seq (bulk-seq) analysis was conducted for two groups harvested on POD 7 (n =3 for each group). Gene enrichment analysis of Ctrl_LW group up-regulated genes (p value < 0.05 and |log₂FoldChange| > 1) illustrated a state readied to incite innate immune responses (Fig. 1i and Supplementary Fig. 2b), indicated by neutrophil chemotaxis and macrophage activation via type 2 immune response. It has been reported that type 2 cytokines such as interleukin 4 receptor, alpha (Il4ra) could activate anti-inflammatory macrophages and lead to the cross-linking of collagen

fibers in scar formation³. The accumulation of type 2 myeloid immune cells in the Ctrl_LW group might be the reason for excessive extracellular matrix organization. In contrast, enrichment analysis of ECM up-regulated genes (p value < 0.05 and $|\log_2\text{FoldChange}| > 1$) revealed enrichment of hair follicle development and mesenchyme morphogenesis driven by Wnt signaling pathway and Hedgehog signaling pathway in the ECM_LW group (Fig. 1i and Supplementary Fig. 2c). The role of Wnt and Hedgehog signaling pathway in regulating T cell development³⁷⁻³⁹ as well as hair follicle regeneration^{40,41} had been stressed. We noticed that the genes such as frizzled class receptor 5 (Fzd5), GATA binding protein 3 (Gata3), and GLI family zinc finger 3 (Gli3) also enriched in T cell differentiation in the thymus. Besides, Gata3 expressed by regulatory T cells (Tregs) are proven to be necessary to prevent excessive collagen deposition driven by type-2 macrophage inflammation⁴², which implicated the importance of adaptive immune system homeostasis in material-primed skin regeneration.

Supplementary Fig. 2 b and c: Chord plots of enriched terms in Ctrl_LW group (b) and ECM_LW group (c).

4) And the same for the enrichment analysis performed using ST (Fig. 3b)

Response: Thanks for the valuable comment. We have supplemented the detailed elaboration of enrichment analysis for differentially expressed genes in ST as suggested. The related descriptions were provided in the Result section of the manuscript with changes marked and the corresponding figure was supplemented in Fig. 3c.

Result (Page 10; Line 236-243): Based on the gene enrichment analysis of ST (Fig. 3b), the Ctrl_LW group enriched the genes related to extracellular matrix organization, collagen biosynthetic process, as well as anti-inflammatory (Mrc1 and Clec10a) and oxidative phosphorylation process (Cox6a2 and Actn3) (Fig. 3c), suggesting the typical AIM-driven fibrous tendency in the Ctrl_LW group. In contrast, the ECM_LW sample supports a metabolic profile associated with the glycolysis process (Eno1, Gapdh). Besides, ECM_LW group expression included gene sets related to the regulation of T cell activation and alpha-beta T cell proliferation that suggested an adaptive immune predominance state in a biomaterial-driven microenvironment (Fig.3 b and c).

Fig. 3c: Violin plot showing the up-regulated genes of Ctrl_LW and ECM_LW samples of ST profile.

5) It's unclear in which analyses you are using only ST data and in which the integrated ST+scRNA-seq data, clearly define it in the text and figures.

Response: Thanks for the valuable comment. For analyses using only ST data (Fig. 3g; Fig. 4e; Fig. 6 d, f, h, and j), the gene's name was displayed in *italics* in the caption of the figures. For analyses using integrated ST+scRNA-seq data (Fig.3 d, e, i, l, m; Fig. 6 c, e, g, i), we have defined it in the text and marked 'integration analysis of xx subclusters' in **bold** in the caption of the figures. The related descriptions were provided in the Result section of the manuscript with changes marked.

Result (Page 11; Line 244-254): To trace the spatial heterogeneity of defined single-cell subclusters, we **integrated the expression profiles of sc-RNA seq and ST using AddmoduleScore function**³². The top 50 (based on avg_log2FC) marker genes of cutaneous cells in the single-cell dataset (defined in Fig. 2) were scored and projected into the ST slices. The spatial feature plot illustrated keratinocytes crawling around the implanted biomaterials (Fig. 3d), which was verified in Supplementary Fig. 2a. Intriguingly, fibroblasts from different lineages showed a distinct spatial distribution pattern in the Ctrl_LW group. Upper lineage PF¹ (HF-related) was limited in the upper layer of granulation tissue, while lower lineage RF¹ (fibrous response-related) predominated in the lower layer (Fig. 3e). In contrast, the distribution of PF¹ (HF-related) was enlarged around biomaterials in the ECM_LW group (Fig. 3e), and ECM_LW showed the higher expression level of PF¹ (Fig. 3f), suggesting the pro-regenerative potential of ECM sample.

Result (Page 11; Line 259-267): To assess the spatial characteristics of scaffold-induced immune microenvironment, we **integrated the immune cells subclusters (defined in Fig. 2) and ST slices in like manner**. Regarding MAC subclusters, ECM showed the ability to reduce the recruitment of AIM¹ (Fig 3 i and j). IF staining of F4/80 and CD206 confirmed the reduced AIM¹ in the ECM_LW group (Fig. 3k), which might relieve the subsequent fibroblast activation and collagen deposition. Instead, we noticed the apparent aggregation of T cells (Fig. 3l), especially Treg¹ (Fig. 3m), colocalized with PF¹ (HF-related) surrounding the biomaterial. The recruitment of Treg¹ (confirmed by IF in Fig. 3n) might contribute to the suppression of the type-2

immunity activated by AIM¹.

Fig. 3: (d) Spatial feature plot showing the distribution of **IFEB¹** and **IFED¹** subclusters in tissue sections. (e) Spatial feature plot showing the distribution of **PF¹** and **RF¹** subclusters in tissue sections.

Fig. 6: (c) Spatial feature plot and violin plot showing the distribution and expression level of **integrated PF²** subcluster in tissue sections. (d) Spatial feature plot and violin plot showing the distribution and expression level of **Crabp1** (marker gene of PF²) in tissue sections

6) Many of the claims in the ST-related section, referring to the visualizations in Fig. 3, need to be supported quantitatively, such as the claims corresponding to Fig. 3c, 3d. Same for 3g,h - some can be clearly supported visually and some cannot and

therefore it's better to add quantitative assessment in a systematic way.

Response: Thanks for the valuable comment. We have supplemented the quantitative assessment of the visualization analysis including fibroblast (Fig. 3f), monocyte and macrophage (Fig. 3j), and T cell subclusters (Fig. 3l and m). Violin plots showed the scores of individual spots derived from scRNA-seq data for each integrated subcluster. Dotted boxes stressed clusters with the higher average scores.

Fig. 3: (f) Violin plots of FIB scores of individual spots derived from scRNA-seq data (sc-FIB score) for each subcluster. Dotted boxes. stressed clusters with the higher average sc-FIB scores. (j) Violin plots of MAC scores of individual spots derived from scRNA-seq data (sc-MAC score) for each subcluster. Dotted boxes stressed clusters with the highest average sc-MAC score. (l) Violin plot showing the expression level of TC¹ in tissue sections. (m) Violin plot showing the expression level of Treg¹ in tissue sections.

7) *Line 192 - you report experimental observations, but the figures you point to (Fig. 4a,b) are schematics.*

Response: Thanks for the valuable comment. We have deleted Fig. 4b and revised the description of experimental observations and schematic figures. The related descriptions were provided in the Result section of the manuscript with changes marked.

Result (Page 12; Line 270-272): Based on the multimodal profiling, we observed a higher proportion of T cells (the schematic summary was shown in Fig. 4a) and HF precursor cells in the ECM_LW group, which implied a potential cellular communication among them.

Fig. 4a: Schematic timeline highlighting the recruitment of immune cells from innate and adaptive immune systems in Ctrl_LW (top) and ECM_LW (bottom) groups.

8) *Lines 194-5 - it sounds from your phrasing that you aim to predict cell-cell communication between the ECM_LW and Ctrl_LW populations (which is not the*

case based on Fig. 4c).

Response: Thanks for the valuable comment. We have revised Fig. 4c and the corresponding description. The related descriptions were provided in the Result section of the manuscript with changes marked and the corresponding figure was revised as Fig. 4b in the revised manuscript.

Result (Page 12; Line 272-280): Next, we applied CellChat to predict the cell-cell communication patterns of immune cells and cutaneous cells between ECM_LW and Ctrl_LW groups. The circle plots showed the overall interaction number of the ligand-receptors in the Ctrl_LW and ECM_LW groups (Fig. 4b). An intensive communication network between fibroblasts and immune cells was observed in both groups. The more abundant interactions in the Ctrl_LW group appeared to be sent from the RF¹, LF¹, MF¹, and Macf¹ subpopulations, indicating the predominance of fibrotic fibroblasts interaction signals in the dermis. In contrast, the more abundant interactions in the ECM_LW group were sent from the Dc¹ and PF¹ subpopulations, indicating the pro-regenerative signals induced by biomaterials.

Fig. 4b: Comparison of overall cell-cell interaction numbers of immune cells and cutaneous cells between Ctrl_LW and ECM_LW using CellChat.

9) Fig. 4c-e should be explained in detail in the caption.

Response: Thanks for the valuable comment. We have revised Fig. 4 c-e, and a detailed description was provided in the caption.

Fig. 4 c-e: (c) The ligand–receptor pairs up-regulated in the Ctrl_LW group in specificity between AIM¹ and fibroblasts (MF¹, RF¹, and LF¹). (d) The ligand–receptor pairs up-regulated in the ECM_LW group in specificity between Treg¹, MAC (Mono¹, PIM¹, AIM¹), and HFSC¹. (e) Spatial feature plots and corresponding violin plots showed the expression level of the ligand and cognate receptor in the Notch signaling pathway.

10) What can we learn from the ligand-receptor analysis using the ST data (e.g. Fig. 4d,e)? Can you see a spatial correlation between the expression of ligands and

receptors? Any other spatially-based conclusions?

Response: Thanks for the valuable comment. We have supplemented the detailed description of ST data (Fig.4 d and e) and added related spatially-based conclusions. The related descriptions were provided in the Result section of the manuscript with changes marked.

Result (Page 12-13; Line 280-293): Since AIM¹ might be the major immune cells contributing to tissue fibrosis, we compared specific interactions among AIM¹ and fibroblast subpopulations (Fig. 4c). The interaction of AIM¹ with fibrotic fibroblasts was more noted in the Ctrl_LW group via Tgfb1–(Tgfbr2+Acvr1b) binding, which play potential pro-fibrotic roles on the target cells¹³. In comparison, the interaction of Treg¹ with HFSC was more noted in the ECM_LW group via Jag1–Notch1 binding, suggesting the pro-regenerative role of Treg¹ in biomaterial-treated wounds⁶⁰. Of note, we detected the more significant interactions of Treg¹ with PIM¹ and Mono¹ via Jag1–Notch2 binding. It has been reported the essential role of Notch signaling in macrophage polarization. Selective inhibitors of Notch signaling significantly suppressed M1-like macrophages and upregulated the M2-like macrophages⁶². Consistent with the gene expression profile at the single-cell level, we confirmed the potential Jag1–Notch2 communication in ST profiles (Fig. 4e). Prominent Notch2 expression was observed around implanted biomaterial, implicating the suppressive role of Tregs in the scaffold-mediated microenvironment.

Fig. 4: Cellular communication landscape between immune cells and cutaneous cells. (c) The ligand–receptor pairs up-regulated in the Ctrl_LW group in specificity between AIM¹ and fibroblasts (MF¹, RF¹, and LF¹). (d) The ligand–receptor pairs up-regulated in the ECM_LW group in specificity between Treg¹, MAC (Mono¹, PIM¹, AIM¹), and HFSC¹. (e) Spatial feature plots and corresponding violin plots showed the expression level of the ligand and cognate receptor in the Notch signaling pathway.

11) Authors need to be more explicit/elaborate when they use terms related to the analysis of the data (e.g. clustering - which type of clustering? what parameters were used? same for integration)

Response: Thanks for the valuable suggestion. Firstly, unsupervised clustering via the FindClusters function (Seurat R packages) was conducted to categorize the cells into clusters based on global gene expression patterns (clustering resolutions of each

dataset were decided by clustree R packages). Secondly, clusters were then assigned to first-level main classes of cells: keratinocytes (KER), fibroblasts (FIB), sebocyte (SEB), smooth muscle cell (SMC), endothelial cell (EC), Schwann cell (SC), melanocyte (MEL), monocyte-macrophage (MAC), T cell (TC), neutrophil (NEU), and dendritic cell (DC). Last but not least, for sub-clustering analysis, we selected cells defined in the first-level clustering and subjected them to a second round of unsupervised clustering (subclustering resolutions of each dataset were decided by clustree R packages).

We have supplemented the methodological details of clustering and integration respectively. The related descriptions were provided in the Result section of the manuscript with changes marked.

Result (Page 6; Line 135-142): At first, to explore the cell composition in the biomaterial-treated wound, we isolated cells from the ECM_LW and Ctrl_LW samples on POD 7, 14, and 21, and applied them to the 10x scRNA-seq platform (Supplementary Fig. 4a). After cell filtering, unsupervised clustering of Seurat categorized the cells into clusters based on global gene expression patterns. Later, clusters were then assigned to first-level main classes of cells. The composition of each main cluster was listed so that the proportion of cells from two groups could be identified across all cell clusters (Supplementary Fig. 4b).

Supplementary Fig. 4: Overview of the single-cell transcriptome analysis between ECM_LW and Ctrl_LW. (a) Single-cell experiment workflow. (b) Cells are categorized into 10 main clusters. The number of cell populations in each cluster, number of cells (%), and composition of ECM_LW and Ctrl_LW groups are listed.

Result (Page 11; Line 244-247): To trace the spatial heterogeneity of defined single-cell subclusters, we integrated the expression profiles of sc-RNA seq and ST using AddmoduleScore function³². The top 50 (based on avg_log2FC) marker genes of cutaneous cells in the single-cell dataset (defined in Fig. 2) were scored and projected into the ST slices.

Fig. 2: The single-cell atlas of the biomaterials-mediated microenvironment. (a) Schematic for generating scRNA-seq and spatial transcriptomics data from large area excisional wounds on POD 7, 14, and 21. (b) Subclustering of keratinocytes showing four subsets from the anagen hair follicle and six subsets from the permanent epidermis. The composition and marker genes for each subset are listed. (c) Subclustering of fibroblasts showing two fibroblast-like subsets and five fibroblast subsets. The marker genes and composition for each subset are listed. (d) Subclustering of monocyte/macrophage showing three subsets. The marker genes, composition, and enrichment analysis for each subset are listed. (e) Subclustering of T cells showing six subsets. The marker genes and composition for each subset are listed.

12) How did the authors batch-correct the single-cell data between the different samples (e.g. ctrl and ECM)

Response: Thanks for the valuable suggestion. We have supplemented the methodological details of the batch-correct process as suggested. The related descriptions were provided in the Online Methods section of the manuscript with changes marked.

Online Methods (Page 18; Line 439-441):

Downstream computational analysis:

Different samples were merged into one Seurat object using the RunHarmony function of harmony R packages (version 0.1.1) to correct the potential batch effect.

13) line 228 - in Fig. 5k there are 8, not 7 clusters shown.

Response: Thanks for the valuable suggestion. We have checked it carefully and revised the correct number as suggested.

14) ST results in Fig. 6 - same comment as above, authors need to support their claims quantitatively (about overall changes in expression and where they occurred in the tissue relative to the wound or relative to other landmarks). And the same for Fig. 7.

Response: Thanks for the valuable suggestion. We have checked it carefully and supplemented the quantitative analysis in Fig.6 and Fig.7 as suggested. The related descriptions were provided in the Result section of the manuscript with changes marked, and the corresponding figures were supplemented in Fig. 6 c-j, Fig. 7f.

Result (Page 14; Line 322-330): Compared to WT samples, the distribution and expression level of PF² (HF-related) was reduced (Fig. 6c), which was verified by the IF staining of the corresponding marker gene Crabp1 (Fig. 6d). In contrast, the distribution and expression level of lower lineage Mest⁺ RF² were significantly increased (Fig. 6 e and f), in accordance with the healing outcome of Rag2^{-/-} groups. To compare the scaffold-induced immune microenvironment, we also integrated marker genes of immune cells (defined in Fig. 5) with ST profiling. We confirmed the more obvious aggregation of AIM² surrounding the biomaterial in the Rag2^{-/-} sample (Fig. 6 g and h), which might contribute to the absence of the Treg² (Fig. 6 i and j).

Result (Page 15; Line 337-341): ST was also applied to explore the spatial characteristics of Ctrl_SW and ECM_SW samples, and the anatomical structure was shown in Fig. 7e. We still observed the recruitment of Cd3⁺ T cells around the implanted biomaterial, which were co-located with Crabp1⁺ papillary fibroblasts in ECM_SW sample (Fig. 7g), confirming the pro-regenerative potential of ECM scaffolds despite of wound size.

Fig. 6 c-j: (c) Spatial feature plot and violin plot showing the distribution and expression level of integrated PF² subcluster in tissue sections. (d) Spatial feature plot and violin plot showing the distribution and expression level of *Crabp1* (marker gene of PF²) in tissue sections; IF staining of CRABP1 (white arrowheads showing the CRABP1⁺ cells). (e) Spatial feature plot and violin plot showing the distribution and expression level of integrated RF² subcluster in tissue sections. (f) Spatial feature plot and violin plot showing the distribution and expression level of *Mest* (marker gene of RF²) in tissue sections; IF staining of MEST (white arrowheads showing the MEST⁺ cells). (g) Spatial feature plot and violin plot showing the distribution and expression level of integrated AIM² subcluster in tissue sections. (h) Spatial feature plot and violin plot showing the distribution and expression level of *Mrc1* (marker gene of AIM²) in tissue sections; IF staining of F4/80 and CD206 (white arrowheads showing the F4/80⁺CD206⁺ cells). (i) Spatial feature plot and violin plot showing the distribution and expression level of integrated Treg² subcluster in tissue sections. (j) Spatial feature plot and violin plot showing the distribution and expression level of *Foxp3* (marker gene of Treg²) in tissue sections; IF staining of FOXP3 (white arrowheads showing the FOXP3⁺ cells).

Fig. 7: (e) The anatomical structure of samples. (f) Spatial feature plot showing the expression of Cd3d (marker gene of T cells) and Crabp1 (marker gene of papillary fibroblasts) in ST profile and corresponding quantitative analysis.

15) Line 253 - the results need to be further elaborated in the text. For example, authors write ‘We further explored the differences in immune cell composition using scRNA-seq and ST (Fig. 7 g, h)’ without any explanation or further discussion of the results shown in these subplots (and the same comment is relevant for almost all results shown for ST analysis related to Fig. 7).

Response: Thanks for the valuable suggestion. Small wounds (6 mm) have faster healing dynamics owing to their much smaller initial size. In reality, 6 mm vs. 1.5 cm wounds will be at functionally very different stages of healing on the same post-wounding day. To some extent, comparing the cell composition of day 7 small vs. day 7 large wounds is improper. We have checked it carefully and deleted the cell composition comparison of ECM_LW and ECM_SW on POD7, and supplemented the evaluation between Ctrl_SW and ECM_SW. Detailed descriptions were supplemented in the Result section, and the corresponding figure was shown in Fig. 7 in the revised manuscript.

Result (Page 14-15; Line 331-341):

Biomaterials facilitate de novo HF regeneration despite wound size

To detect any differences in wound healing between small and large wounds treated with biomaterials, we repeated the experiment with small (diameter=0.6 cm) full-thickness wounds implanted with biomaterials (Fig. 7 a and b). Consistent with the ECM_LW group, the scaffold implanted in the small wound (ECM_SW) did not trigger an obvious FBR fibrous capsule and had a rapid degradation rate (Fig. 7c). Histological sections of ECM_SW tissue revealed clear signs of enhanced HF reconstruction too (Fig. 7 c and d). ST was also applied to explore the spatial characteristics of Ctrl_SW and ECM_SW samples, and the anatomical structure was shown in Fig. 7e. We still observed the recruitment of Cd3⁺ T cells around the implanted biomaterial, which were co-located with Crabp1⁺ papillary fibroblasts in ECM_SW sample (Fig. 7g), confirming the pro-regenerative potential of ECM scaffolds despite of wound size.

Fig. 7: (a) Workflow for evaluating skin wound healing. (b) The surgical process for skin excisional wound model of Ctrl_SW and ECM_SW group. (c) H&E staining of Ctrl_SW and ECM_SW samples. (d) Histologic quantification of de novo HF on POD28 (n=5 for each group). (e) The anatomical structure of samples. (f) Spatial feature plot showing the expression of Cd3d (marker gene of T cells) and Crabp1 (marker gene of papillary fibroblasts) in ST profile and corresponding quantitative analysis. ***P < 0.0001, **P < 0.01, and *P < 0.05 by Student's t-test for data in (d).

16) In the methods section the authors should dedicate additional subsections to the computational analysis of the data

Response: Thanks for the valuable suggestion. We have supplemented the additional subsections to the computational analysis of the data as suggested. The related descriptions were provided in the Online Methods section of the manuscript with changes marked.

Online Methods (Page 18-19; Line 420-480):

Single cell RNA sequencing

Tissue Dissociation

Sequencing and data processing

Downstream computational analysis

Spatial Transcriptomics

Slide preparation

Downstream computational analysis

Integration analysis of scRNA-seq and ST

Minor comments:

1) many of the font sizes in the figures are too small to read

Response: Thanks for the valuable suggestion. We have checked it carefully and adjusted the font sizes in the figures as suggested.

2) the legend in fig. 1i is unclear

Response: Thanks for the valuable suggestion. We have checked it carefully and supplemented the detailed elaborations as suggested (see the words marked with highlights in the figure legends of Fig. 1i).

Fig. 1i: Bulk-RNA sequencing analysis of ECM_LW versus Ctrl_LW mice on POD7 (n=3 for each group). Heatmap (left) showing hierarchical clustering of differentially

expressed genes (p value<0.05 & |log2FC|> 1) between two groups, and corresponding gene set enrichment analysis (right) showing the enriched terms in ECM_LW (top) versus Ctrl_LW (bottom) groups.

3) line 195: typo in ECM

Response: Thanks for the valuable suggestion. We have checked it carefully and revised it into 'ECM_LW' as suggested (Line 273 in the revised manuscript).

We sincerely thank **Reviewer 1** for reviewing our manuscript and providing meaningful comments so we can significantly improve our research.

Referee 2

Reviewer #2 (Remarks to the Author):

Skin scarring without dermal appendage formation is a common problem in wound healing. The manuscript deals with this by characterizing the effect of a biodegradable aligned extracellular matrix (ECM) scaffold. They observed faster wound closure and enhanced hair follicle neogenesis by the implantation of ECM scaffold in mouse excisional wounds. This study then utilized single-cell RNA sequencing and spatial transcriptomics to understand the effect of biodegradable ECM scaffold during wound healing with a focus on immune cell populations and hair follicle neogenesis. They also showed that immunodeficient mice (Rag2 -/-) show defects in hair follicle neogenesis. However, the project is very descriptive. As scRNA seq of hair follicle neogenesis has been published by several groups, the novelty of this study is limited. More specific and in-depth understanding of how components of ECM bring about cellular and molecular changes would be necessary to establish a solid novel model/theory in this field.

Response: We appreciate Reviewer 2 for carefully reading our manuscript and making insightful, critical, and constructive feedback, which has enabled us to prepare a greatly improved manuscript. We apologize for the previous poor formatting and some disputed descriptions of our manuscript, which may have made some of this evidence unclear. We have supplemented the specific and in-depth analysis of multimodal data, which highlights the essential role of regulatory T cells in mitigating tissue fibrosis by suppressing excessive type 2 inflammation in the ECM scaffold-mediated microenvironment. And we have revised the statements in our abstract, introduction, result, and discussion to better highlight the novelty and advancement of our work. Below, we address the questions and suggestions raised by Reviewer 2 point-by-point.

1) Several studies indicate the incidence and extent of hair follicle neogenesis vary between samples. Quantifications for hair follicle regeneration need to be more rigorous. What percentage of the samples contained de novo hair follicle in each

condition? Picture of healing time course and hair follicle neogenesis in control without splint needs to be demonstrated.

Response: Thanks for the valuable suggestion. We have revised the figures of quantifications for hair follicles, showing the individual values of each sample as suggested. The revised figures were provided in Fig. 1h, Fig. 7d, and Supplementary Fig. 1e. In addition, we have supplemented the healing time course and hair follicle neogenesis evaluation in control without splint as suggested. The related descriptions were shown in the Result section with changes marked, and the corresponding figure was shown in Supplementary Fig.1.

Result (Page 4; Line 85-95): C57BL/6 mice are a classical choice for studying wound healing due to their accessibility, affordability, and ease of handling. It is worth noting that wound healing in rodent models primarily relies on contraction by the panniculus carnosus for wound closure, whereas in humans, re-epithelialization and granulation tissue formation play a larger role³⁶. As shown in Supplementary Fig. 1a, we created a splinted wound excisional model that could restrict the contraction of the panniculus carnosus, while the unsplinted group was treated without a silicone splint. The decreased rate of wound closure (Supplementary Fig. 1b) and the increased granulation formation and de novo HF were observed in the splinted mice (Supplementary Fig. 1 c-e). In order to minimize wound contraction in rodents and mimic the wound healing process in humans with tight skin, we chose the splinted wound healing model for further evaluation.

Fig. 1h: Histologic quantification of de novo HF on POD28 (n=5 for each group)

Fig. 7d: Histologic quantification of de novo HF's on POD28 (n=5 for each group)

Supplementary Fig. 1e: (e) Histologic quantification of de novo HF's on POD28 (n=5 for each group).

Supplementary Fig. 1: Evaluation of the healing process of splinted and unsplinted wounds. (a) Residual wound area of splinted and unsplinted wounds treated with saline or ECM scaffolds, and (b) corresponding analysis (n=4 for each group). (c) H&E analysis of each group on POD 7 and POD28. (d) Quantitative evaluation of the area of granulation tissue (n=5 for each group). (e) Histologic quantification of de novo HFs on POD28 (n=5 for each group). ***P < 0.0001, **P < 0.01, and *P < 0.05 by ANOVA analysis for data in (b) and Student's t-test for data in (d) & (e).

2) The reasoning for examination of Day 7 post wounding in scRNA seq analyses is insufficient. ECM may change the cell composition at other time points and influence the cell state of wound healing.

Response: Thanks for the valuable suggestion. We have supplemented the scRNA analysis of Ctrl_LW and ECM_LW groups harvested on POD 14 (ECM_LW: 13985 cells / Ctrl_LW:13129 cells) and POD 21 (ECM_LW: 7115 cells / Ctrl_LW:10279 cells) as suggested. By analyzing scRNA data at various time points, we discovered that ECM scaffolds have the ability to consistently attract immunosuppressive Treg cells while simultaneously reducing type 2 immune cells. The detailed analysis was revised in the Result and Discussion section with change marked, and corresponding figures were supplemented in Supplementary Fig. 4 and Fig.2 (highlighted with red box).

Result (Page 6; Line 138-142): At first, to explore the cell composition in the biomaterial-treated wound, we isolated cells from the ECM_LW and Ctrl_LW samples on POD 7, 14, and 21, and applied them to the 10x scRNA-seq platform (Supplementary Fig. 4a). After cell filtering, unsupervised clustering of Seurat categorized the cells into clusters based on global gene expression patterns. Later, clusters were then assigned to first-level main classes of cells. The composition of each main cluster was listed so that the proportion of cells from two groups could be identified across all cell clusters (Supplementary Fig. 4b).

Supplementary Fig. 4: Overview of the single-cell transcriptome analysis between ECM_LW and Ctrl_LW. (a) Single-cell experiment workflow. (b) Cells are categorized into 10 main clusters. The number of cell populations in each cluster, number of cells (%), and composition of ECM_LW and Ctrl_LW groups are listed.

Result (Page 7; Line 145-157): Firstly, we selected main clusters defined as keratinocytes and subjected them to a second round of unsupervised clustering (Fig. 2b). The heterogeneity of keratinocyte subclusters of this dataset corresponded with the healing outcomes of two groups: the higher proportion of Krt5⁺ interfollicular epidermal basal cell¹ (IFEB¹) and Krt10⁺ interfollicular epidermal differentiated cell¹ (IFED¹) were observed in the ECM_LW group on **POD7**, supporting more neo-epithelium proliferation in the presence of scaffolds in the proliferative stage (Supplementary Fig. 2a). HF neogenesis was believed to occur through the migration of epithelial HFSC or hair follicle progenitor (HFP) to the wound center and form the placodes to activate papillary fibroblast (PF) fate specification into dermal condensate (Dc)^{40,43,44}. In accordance with reports, we found a higher proportion of Krt25⁺Krt28⁺Krt17⁺ hair follicle progenitor¹ (HFP¹)²² and Crip1⁺Lrig1⁺Lgr5⁺ hair follicle stem cell¹ (HFSC¹)⁵ in ECM_LW group on **POD7, 14 and 21** (Fig. 2b), which might serve enough epithelial resources for the following nascent HFs reconstruction.

Result (Page 8; Line 172-181): In accordance with previous report^{43,51}, the primary wave of dermal repair in the Ctrl_LW group was mediated by the lower lineage fibroblast including Gpx3⁺Mest⁺ reticular fibroblast¹ (RF¹) and Mfap4⁺ Cd34⁺ lipofibroblast¹ (LF¹) on **POD7 and POD14**, which was respectively related to dermis collagen fibril organization and hypodermis adipocytes formation in GO enrichment analysis (Supplementary Fig. 5b). It's worth noting that RF¹ highly expressed genes enriched in innate immune system, in which Il33⁵², I4ra¹⁴ and Il13ra1⁹ were related to the initiation of type 2 macrophage inflammation and collagen deposition in fibrous disease. In contrast, there were more upper lineage Crabp1⁺Prdm1⁺PF¹ in biomaterial-implanted wounds, which was believed to have the capacity to support HF initiation^{48,53} (Supplementary Fig. 5b).

Result (Page 9-10; Line 199-232): Monocytes and macrophages (MAC) were vital in biomaterial-related FBR⁵⁵, orchestrating tissue repair by modulating fibroblast activation⁹. Three subsets of MAC, including Cd14⁺Ly6c2⁺ monocytes¹ (Mono¹), Ptgs2⁺Nos2⁺ pro-inflammatory macrophages¹ (PIM¹), and Mrc1⁺ C1qa⁺ anti-inflammatory macrophages¹ (AIM¹), were determined (Fig. 2d). As shown in enrichment analysis, pro-inflammatory (classically /M1-like) macrophages can be activated from monocyte by a variety of alarmins, which result in the expression of pro-inflammatory cytokines including TNF and IL-1 β . Induced by the IL-4 and IL-13, anti-inflammatory (alternatively activated /M2-like) subsets are the foremost effectors of type 2 immune response, which play an essential role in aberrant collagen deposition¹⁴ through the expression of TGF- β and various matrix remodeling mediators⁵⁶. Of note, AIM¹, which enriched genes related to collagen formation, was the only subcluster that continuously decreased in the ECM_LW group on **POD 7, 14, and 21**, suggesting that the scaffold might play a role in reducing type 2 macrophage infiltration.

Although T cells are not required for wound healing, they are capable of modulating each phase of tissue repair by regulating macrophage activity. T cells were the only immune cell that owned a higher proportion in the ECM_LW group (Supplementary Fig. 4b), and genes related to T cells were substantially higher expressed by the

ECM_LW group (Supplementary Fig. 4d), suggesting the activation of adaptive immune system induced by biomaterials. Subclustering of T cells resulted in six subsets including Cd8b1⁺ cytotoxic T cell¹ (CTL¹), Tyrobp⁺Nkg7⁺Xcl⁺ nature killer cell¹ (NKT¹), Il17a⁺Il17f⁺ T helper17¹ (Th17¹), Foxp3⁺Gata3⁺ T regulatory cell¹ (Treg¹), Trdv4⁺Cd7⁺ $\gamma\delta$ T cell¹ (T $\gamma\delta$ ¹) and Gata3⁺Il4⁺ T helper2¹ (Th2¹) based on markers from published research^{57,58}. More cells in the Th2¹ populations were from the Ctrl_LW samples **on POD7, 14, and 21**, whereas the larger number of Treg¹ were from the ECM_LW samples. Il4⁺ Il13⁺ Th2 cells are one of the effector cells associated with type 2 immune responses, which could facilitate the differentiation of type-2 AIM⁵². Located in the dermis near HF in normal skin, regulatory T cells (Tregs) are believed to maintain the balance of immune homeostasis. It has been reported that Gata3⁺ Tregs could restrain Th2-mediated fibroblast activation and scarring in murine cutaneous fibrosis⁴², and clinical IL-2 therapy (which facilitates Treg proliferation and activation) has also been proven to be effective in alleviating chronic skin fibrosis⁵⁹. In addition, Tregs could also facilitate HFSC differentiation to initiate HF regeneration through Jag1-Notch1 pathways⁶⁰. Gene enrichment analysis confirmed that the Treg¹ subpopulation was enriched in signaling pathways regulating pluripotency of stem cells (Supplementary Fig. 5c), suggesting that Treg¹ recruited by ECM scaffolds might initiate HF neogenesis by both immune-suppression and pro-reparative functions⁶¹.

Fig. 2: The single-cell atlas of the biomaterials-mediated microenvironment. (b) Subclustering of keratinocytes showing four subsets from the anagen hair follicle and six subsets from the permanent epidermis. The composition and marker genes for each subset are listed. (c) Subclustering of fibroblasts showing two fibroblast-like subsets and five fibroblast subsets. The marker genes and composition for each subset are listed. (d) Subclustering of monocyte/macrophage showing three subsets. The marker genes, composition, and enrichment analysis for each subset are listed. (e) Subclustering of T cells showing six subsets. The marker genes and composition for each subset are listed.

Discussion (Page 15-16; Line 354-361): By multimodal analysis, we observed the substantial accumulation of type 2 immune cells (especially Mrc1⁺ AIM and Gata3⁺ Th2 cells) in Ctrl_LW samples (Fig. 2 d and e), which might promote the fast healing

of damaged tissue at the expense of the original skin composition and function^{67,68}. In contrast, adaptive T cell infiltration in response to scaffold implantation was driven towards immunosuppression subpopulations in the ECM_LW group. A larger number of Foxp3⁺ Tregs were recruited by ECM scaffold to mitigate skin fibrosis by suppressing excessive type 2 macrophage inflammation (Fig. 2e).

3) *If omics suggest hair neogenesis occurs on day 7 prior to re-epithelialization with ECM biomaterials, such examples need to be verified by IF or histology.*

Response: Thanks for the valuable suggestion. Considering the KRT28 antibody (reactivity: mouse) for IF staining was not available, we have supplemented the IF staining of hair germ (KRT17/ green) and dermal condensate (TWIST2/ red) to verify the presence of primary hair follicle structure in ECM_LW group on POD7 (Fig. 1g).

Result (Page 5; Line 103-107): The nascent HF^s induced by ECM scaffolds mimicked embryonic hair follicle development pattern (Fig. 1g), with high KRT17 (green) in hair germ (HG) and TWIST2 (red) in dermal condensate (Dc) on POD7. After morphogenesis, neogenic HF^s in the ECM_LW group contained proliferating epithelial cells expressing Ki67 (red) with sebaceous glands (SCD1/ green).

Fig. 1g: Nascent HF^s within the ECM_LW group were stained for KRT17 (green) and TWIST2 (red), Ki67 (red), and SCD1 (green). Abbreviations: HF, hair follicle; HG, hair germ; Dc, dermal condensate; SG, sebaceous gland.

4) *How does the ECM scaffold bring about cellular and molecular changes?*

Specific mechanisms of how ECM regulates specific immune cells need to be analyzed in association with functional data.

Response: Thanks for the valuable suggestion. Multimodal analysis (ECM_LW vs. Ctrl_LW) highlighted the potential role of regulatory T cells recruited by ECM scaffolds in mitigating tissue fibrosis by suppressing excessive type 2 inflammation. Immunodeficient mice lacking mature T lymphocytes (Rag2^{-/-} vs. WT) showed the typical characteristic of tissue fibrosis driven by type 2 macrophage inflammation, validating the potential therapeutic effect of the adaptive immune system activated by biomaterials. We have supplemented the specific mechanisms of how ECM regulates specific immune cells as suggested. The related descriptions were supplemented in the Abstract, Introduction, Result, and Discussion sections with changes marked.

Abstract (Page 1; Line 20-26): Multimodal analysis, in combination with single-cell RNA sequencing and spatial transcriptomics, provided a unique lens through which we can explore the delicate immune responses around biomaterials, highlighting the potential role of regulatory T cells in mitigating tissue fibrosis by suppressing excessive type 2 inflammation. Immunodeficient mice lacking mature T lymphocytes showed the typical characteristic of tissue fibrosis driven by type 2 macrophage inflammation, validating the potential therapeutic effect of the adaptive immune system activated by biomaterials.

Introduction (Page 2-3; Line 33-53): The immune system plays a varying role in driving scar fibrosis²⁻⁴ or HF regeneration⁵⁻⁸ upon different environmental stimuli. Macrophages contribute to all phases of tissue repair, and the heterogeneity of macrophages is believed to be one of the critical orchestrators determining the healing outcome^{9,10}. Two major subpopulations, including pro-inflammatory M1-like and anti-inflammatory M2-like, have been credited with these distinct roles¹¹. It has been reported that the pro-inflammatory macrophage-elicited pro-inflammatory mediators, such as tumor necrosis factor (TNF)⁵ and interleukin-1 beta (IL-1 β)¹², effectively promote subsequent HF neogenesis. In contrast, type-2 anti-inflammatory macrophages might play an essential role in wound fibrosis by promoting fibrotic

fibroblast activation and collagen cross-linking³ via fibrotic cytokines such as transforming growth factor- β (TGF- β)¹³ and RELM α ¹⁴ or chronic phagocytosis activity⁴ at a later stage. Although the role of macrophages in pathogen clearance and tissue fibrosis¹⁵ has long been stressed, only recently have the T lymphocytes been more thoroughly investigated. In addition to $\gamma\delta$ T cells that induce HF neogenesis through the secretion of fibroblast growth factor 9 (Fgf9)⁶, the adaptive T cells are being explored to gain more understanding of its role in regulating macrophage polarization and, thus, wound regeneration². T cells coordinate the polarized immune responses through differentiation into specialized subsets of helper T cells (Th1, Th2, and Th17) and drive the type 1/2/3 paradigm of immunity¹⁶. Besides, specialized regulatory T cells (Tregs) have evolved to counterbalance the potentially detrimental effect of the innate immune system by suppressing macrophage response¹⁷ and facilitating wound regeneration.

Result (Page 5; Line 112-132): To explore the underlying mechanism, a bulk-tissue RNA-seq (bulk-seq) analysis was conducted for two groups harvested on day 7 (n =3 for each group). Gene enrichment analysis of Ctrl_LW group up-regulated genes (p value < 0.05 and $|\log_2\text{FoldChange}| > 1$) illustrated a state readied to incite innate immune responses (Fig. 1i and Supplementary Fig. 2b), indicated by neutrophil chemotaxis and macrophage activation via type 2 immune response. It has been reported that type 2 cytokines such as interleukin 4 receptor, alpha (Il4ra) could activate anti-inflammatory macrophages and lead to the cross-linking of collagen fibers in scar formation³. The accumulation of type 2 myeloid immune cells in the Ctrl_LW group might be the reason for excessive extracellular matrix organization. In contrast, enrichment analysis of ECM up-regulated genes (p value < 0.05 and $|\log_2\text{FoldChange}| > 1$) revealed enrichment of hair follicle development and mesenchyme morphogenesis driven by Wnt signaling pathway and Hedgehog signaling pathway in the ECM_LW group (Fig. 1i and Supplementary Fig. 2c). The role of Wnt and Hedgehog signaling pathway in regulating T cell development³⁷⁻³⁹ as well as hair follicle regeneration^{40,41} had been stressed. We noticed that the genes such

as frizzled class receptor 5 (Fzd5), GATA binding protein 3 (Gata3), and GLI family zinc finger 3 (Gli3) also enriched in T cell differentiation in the thymus. Besides, Gata3 expressed by regulatory T cells (Tregs) are proven to be necessary to prevent excessive collagen deposition driven by type-2 macrophage inflammation⁴², which implicated the importance of adaptive immune system homeostasis in material-primed skin regeneration. Flow cytometry (Fig. 1j and Supplementary Fig. 3) and immunohistochemistry (IHC) staining (Supplementary Fig. 2d) confirmed the increased T cell (CD3⁺) infiltration around implanted biomaterials in the early phase.

Fig. 1: (i) Bulk-RNA sequencing analysis of ECM_LW versus Ctrl_LW mice on POD7 (n=3 for each group). Heatmap (left) showing hierarchical clustering of differentially expressed genes (p value<0.05 & |log2FC|> 1) between two groups, and corresponding gene set enrichment analysis (right) showing the enriched terms in ECM_LW (top) versus Ctrl_LW (bottom) groups. (j) Proportions of T cells (CD45⁺CD3⁺) and macrophages (CD45⁺ CD3⁻F4/80⁺CD68⁺) cell populations in the wound environment on POD7, determined by flow cytometry (% = the number of target cells / the number of all single live cells) (n=3 for each group).

Supplementary Fig. 2 : (b-c) Chord plots of enriched terms in Ctrl_LW group (b) and ECM_LW group (c). (d) IHC staining of CD3 and CD68 on POD7 and corresponding quantitative analysis (n=5 for each group).

Supplementary Fig. 3: Gating scheme for T cells and macrophages. Multicolor flow cytometry gating strategy to isolate T cells (CD45⁺CD3⁺) and macrophages (CD45⁺CD3⁻ CD68⁺F4/80⁺) from single cell suspension of Ctrl_LW (a) and ECM_LW (b) samples on POD7.

Result (Page 8; Line 172-179): In accordance with previous report^{43,51}, the primary wave of dermal repair in the Ctrl_LW group was mediated by the lower lineage fibroblast including Gpx3⁺Mest⁺ reticular fibroblast¹ (RF¹) and Mfap4⁺ Cd34⁺ lipo-fibroblast¹ (LF¹) on POD7 and POD14, which was respectively related to dermis collagen fibril organization and hypodermis adipocytes formation in GO enrichment analysis (Supplementary Fig. 5b). It's worth noting that RF¹ highly expressed genes enriched in innate immune system, in which Il33⁵², I4ra¹⁴ and Il13ra1⁹ were related to the initiation of type 2 macrophage inflammation and collagen deposition in fibrous disease.

Supplementary Fig. 5b: GO enrichment analysis of fibroblast subtypes.

Result (Page 9; Line 199-211): Monocytes and macrophages (MAC) were vital in biomaterial-related FBR⁵⁵, orchestrating tissue repair by modulating fibroblast activation⁹. Three subsets of MAC, including Cd14⁺Ly6c2⁺ monocytes¹ (Mono¹),

Ptgs2⁺Nos2⁺ pro-inflammatory macrophages¹ (PIM¹), and Mrc1⁺ C1qa⁺ anti-inflammatory macrophages¹ (AIM¹), were determined (Fig. 2d). As shown in enrichment analysis, pro-inflammatory (classically /M1-like) macrophages can be activated from monocyte by a variety of alarmins, which result in the expression of pro-inflammatory cytokines including TNF and IL-1 β . Induced by the IL-4 and IL-13, anti-inflammatory (alternatively activated /M2-like) subsets are the foremost effectors of type 2 immune response, which play an essential role in aberrant collagen deposition¹⁴ through the expression of TGF- β and various matrix remodeling mediators⁵⁶. Of note, AIM¹, which enriched genes related to collagen formation, was the only subcluster that continuously decreased in the ECM_LW group on POD 7, 14, and 21, suggesting that the scaffold might play a role in reducing type 2 macrophage infiltration.

Fig. 2d: The single-cell atlas of the biomaterials-mediated microenvironment. (d) Subclustering of monocyte/macrophage showing three subsets. The marker genes, composition, and enrichment analysis for each subset are listed.

Result (Page 10; Line 220-232): More cells in the Th2¹ populations were from the Ctrl_LW samples on POD7, 14, and 21, whereas the larger number of Treg¹ were from the ECM_LW samples (Fig. 2e). Il4⁺ Il13⁺ Th2 cells are one of the effector cells associated with type 2 immune responses, which could facilitate the differentiation of type-2 AIM⁵². Located in the dermis near HF in normal skin, regulatory T cells (Tregs) are believed to maintain the balance of immune homeostasis. It has been reported that Gata3⁺ Tregs could restrain Th2-mediated fibroblast activation and scarring in murine cutaneous fibrosis⁴², and clinical IL-2 therapy (which facilitates Treg proliferation and activation) has also been proven to be effective in alleviating

chronic skin fibrosis⁵⁹. In addition, Tregs could also facilitate HFSC differentiation to initiate HF regeneration through Jag1-Notch1 pathways⁶⁰. Gene enrichment analysis confirmed that the Treg¹ subpopulation was enriched in signaling pathways regulating pluripotency of stem cells (Supplementary Fig. 5c), suggesting that Treg¹ recruited by ECM scaffolds might initiate HF neogenesis by both immune-suppression and pro-reparative functions⁶¹.

Fig. 2e: The single-cell atlas of the biomaterials-mediated microenvironment. (e) Subclustering of T cells showing six subsets. The marker genes and composition for each subset are listed.

Supplementary Fig. 5c: Enrichment analysis of T cell subtypes.

Result (Page 11-12; Line 259-267): To assess the spatial characteristics of scaffold-induced immune microenvironment, we integrated the immune cells subclusters (defined in Fig. 2) and ST slices in like manner. Regarding MAC subclusters, ECM showed the ability to reduce the recruitment of AIM¹ (Fig 3 i and j). IF staining of F4/80 and CD206 confirmed the reduced AIM¹ in the ECM_LW group (Fig. 3k), which might relieve the subsequent fibroblast activation and collagen deposition.

Instead, we noticed the apparent aggregation of T cells (Fig. 3l), especially Treg¹ (Fig. 3m), colocalized with PF¹ (HF-related) surrounding the biomaterial. The recruitment of Treg¹ (confirmed by IF in Fig. 3n) might contribute to the suppression of the type-2 immunity activated by AIM¹.

Fig. 2: The single-cell atlas of the biomaterials-mediated microenvironment. (a)

Schematic for generating scRNA-seq and spatial transcriptomics data from large area excisional wounds on POD 7, 14, and 21. (b) Subclustering of keratinocytes showing four subsets from the anagen hair follicle and six subsets from the permanent epidermis. The composition and marker genes for each subset are listed. (c) Subclustering of fibroblasts showing two fibroblast-like subsets and five fibroblast subsets. The marker genes and composition for each subset are listed. (d)

Subclustering of monocyte/macrophage showing three subsets. The marker genes, composition, and enrichment analysis for each subset are listed. (e) Subclustering of T cells showing six subsets. The marker genes and composition for each subset are listed.

Fig. 3: Spatial anchors tracing the cell distribution around the ECM scaffold. (i) Spatial feature plot showing the distribution of MAC subclusters in tissue sections. (j) Violin plots of MAC scores of individual spots derived from scRNA-seq data (sc-MAC score) for each subcluster. Dotted boxes stressed clusters with the highest average sc-MAC score. (k) IF staining of F4/80 and CD206 (white arrowheads showing the F4/80⁺CD206⁺ cells). (l) Spatial feature plot and violin plot showing TC¹ distribution and expression level in tissue sections. (m) Spatial feature plot and violin plot showing the distribution and expression level of Treg¹ in tissue sections. (n) IF staining of FOXP3 (white arrowheads showing the FOXP3⁺ cells).

Result (Page 12-13; Line 280-293): Since AIM¹ might be the major immune cells contributing to tissue fibrosis, we compared specific interactions among AIM¹ and fibroblast subpopulations (Fig. 4c). The interaction of AIM¹ with fibrotic fibroblasts was more noted in the Ctrl_LW group via Tgfb1–(Tgfr2+Acvr1b) binding, which play potential pro-fibrotic roles on the target cells¹³. In comparison, the interaction of

Treg¹ with HFSC was more noted in the ECM_LW group via Jag1–Notch1 binding, suggesting the pro-regenerative role of Treg¹ in biomaterial-treated wounds⁶⁰. Of note, we detected the more significant interactions of Treg¹ with PIM¹ and Mono¹ via Jag1–Notch2 binding. It has been reported the essential role of Notch signaling in macrophage polarization. Selective inhibitors of Notch signaling significantly suppressed M1-like macrophages and upregulated the M2-like macrophages⁶². Consistent with the gene expression profile at the single-cell level, we confirmed the potential Jag1–Notch2 communication in ST profiles (Fig. 4e). Prominent Notch2 expression was observed around implanted biomaterial, implicating the suppressive role of Tregs in the scaffold-mediated microenvironment.

Fig. 4: Cellular communication landscape between immune cells and cutaneous cells.

(c) The ligand–receptor pairs up-regulated in the Ctrl_LW group in specificity between AIM¹ and fibroblasts (MF¹, RF¹, and LF¹). (d) The ligand–receptor pairs up-regulated in the ECM_LW group in specificity between Treg¹, MAC (Mono¹, PIM¹, AIM¹), and HFSC¹. (e) Spatial feature plots and corresponding violin plots showed

the expression level of the ligand and cognate receptor in the Notch signaling pathway.

Result (Page 13-14; Line 310-317): Subclustering of MAC resulted in three subsets including PIM², AIM², and Mono² (Fig. 5g). Rag2^{-/-} samples improved the recruitment of AIM², PIM², in which AIM² owned the highest proportion (46.2%). Subclustering of T cells resulted in seven subsets (Fig. 5h). The Rag2^{-/-} groups contributed to fewer T cells in most subsets except NKT² (related to natural killer cell mediated cytotoxicity) and naive Th2², indicating the dysfunctional adaptive immune systems in Rag2^{-/-} group. The absence of suppressive Treg² might be the reason for uncontrolled type-2 AIM² accumulation and collagen deposition of fibrotic RF².

Fig. 5: Evaluation of wound healing in immunodeficient mice lacking mature T cells. (g) Subclustering of monocyte/macrophage showing three subsets. The marker genes, composition, and enrichment analysis for each subset are listed. (h) Subclustering of T cells showing seven subsets. The marker genes and composition for each subset are listed.

Result (Page 14; Line 322-330): Compared to WT samples, the distribution and expression level of PF² (HF-related) was reduced (Fig. 6c), which was verified by the IF staining of the corresponding marker gene Crabp1(Fig. 6d). In contrast, the distribution and expression level of lower lineage Mest⁺ RF² were significantly

increased (Fig. 6 e and f), in accordance with the healing outcome of Rag2^{-/-} groups. To compare the scaffold-induced immune microenvironment, we also integrated marker genes of immune cells (defined in Fig. 5) with ST profiling. We confirmed the more obvious aggregation of AIM² surrounding the biomaterial in the Rag2^{-/-} sample (Fig. 6 g and h), which might contribute to the absence of the Treg² (Fig. 6 i and j).

Fig. 5: Evaluation of wound healing in immunodeficient mice lacking mature T cells.

(a) The surgical process for evaluating large area wound healing mediated by ECM scaffold in WT and Rag2^{-/-} mice. (b) Histological analysis of wound healing in WT and Rag2^{-/-} mice at 7 and 28 days. (c) Residual defect area on POD 7 (n=3 for each group). (d) Semiquantitative evaluation of gap width (n=3 for each group). (e)

Histologic quantification of de novo HF^s (n=5 for each group). (f) Subclustering of fibroblasts and fibroblast-like cells showing two fibroblast-like subsets and five fibroblast subsets. Marker genes for fibroblast subsets are listed. (g) Subclustering of monocyte/macrophage showing three subsets. The marker genes, composition, and enrichment analysis for each subset are listed. (h) Subclustering of T cells showing seven subsets. The marker genes and composition for each subset are listed. ***P < 0.0001, **P < 0.01, and *P < 0.05 by Student's t-test for data in (c), (d), and (e).

Fig. 6: Spatial atlas of cell microenvironment around biomaterials of immunodeficient mice. (a) The anatomical structure of each sample. (b) Gene enrichment analysis between WT and Rag2^{-/-} group. (c) Spatial feature plot and violin plot showing the distribution and expression level of integrated PF² subcluster in tissue sections. (d) Spatial feature plot and violin plot showing the distribution and expression level of *Crabp1* (marker gene of PF²) in tissue sections; IF staining of CRABP1 (white arrowheads showing the CRABP1⁺ cells). (e) Spatial feature plot and violin plot showing the distribution and expression level of integrated RF² subcluster in tissue sections. (f) Spatial feature plot and violin plot showing the distribution and expression level of *Mest* (marker gene of RF²) in tissue sections; IF staining of MEST

(white arrowheads showing the MEST⁺ cells). (g) Spatial feature plot and violin plot showing the distribution and expression level of integrated AIM² subcluster in tissue sections. (h) Spatial feature plot and violin plot showing the distribution and expression level of Mrc1 (marker gene of AIM²) in tissue sections; IF staining of F4/80 and CD206 (white arrowheads showing the F4/80⁺CD206⁺ cells). (i) Spatial feature plot and violin plot showing the distribution and expression level of integrated Treg² subcluster in tissue sections. (j) Spatial feature plot and violin plot showing the distribution and expression level of Foxp3 (marker gene of Treg²) in tissue sections; IF staining of FOXP3 (white arrowheads showing the FOXP3⁺ cells).

Discussion (Page 15-16; Line 354-365): By multimodal analysis, we observed the substantial accumulation of type 2 immune cells (especially Mrc1⁺ AIM and Gata3⁺ Th2 cells) in Ctrl_LW samples (Fig. 2 d and e), which might promote the fast healing of damaged tissue at the expense of the original skin composition and function^{67,68}. In contrast, adaptive T cell infiltration in response to scaffold implantation was driven towards immunosuppression subpopulations in the ECM_LW group. A larger number of Foxp3⁺ Tregs were recruited by ECM scaffold to mitigate skin fibrosis by suppressing excessive type 2 macrophage inflammation (Fig. 2e and 3l-3m). We next confirmed the requirement of T cells in skin regeneration by an immunodeficient model (Fig. 5). The absence of suppressive Treg² might be one of the reasons for the uncontrolled accumulation of type-2 AIM², which might drive collagen deposition by activating profibrotic RF². These data validated that the activation of adaptive immune was required for the reparative effect mediated by ECM scaffolds.

Fig. 2 d and e: The single-cell atlas of the biomaterials-mediated microenvironment.

(d) Subclustering of monocyte/macrophage showing three subsets. The marker genes, composition, and enrichment analysis for each subset are listed. (e) Subclustering of T cells showing six subsets. The marker genes and composition for each subset are listed.

Fig. 3 l and m: Spatial anchors tracing the cell distribution around the ECM scaffold.

(l) Spatial feature plot and violin plot showing TC¹ distribution and expression level in tissue sections. (m) Spatial feature plot and violin plot showing the distribution and expression level of Treg¹ in tissue sections.

Fig. 5: Evaluation of wound healing in immunodeficient mice lacking mature T cells. (a) The surgical process for evaluating large area wound healing mediated by ECM scaffold in WT and Rag2^{-/-} mice. (b) Histological analysis of wound healing in WT and Rag2^{-/-} mice at 7 and 28 days. (c) Residual defect area on POD 7 (n=3 for each group). (d) Semiquantitative evaluation of gap width (n=3 for each group). (e) Histologic quantification of de novo HFs (n=5 for each group). (f) Subclustering of fibroblasts and fibroblast-like cells showing two fibroblast-like subsets and five fibroblast subsets. Marker genes for fibroblast subsets are listed. (g) Subclustering of monocyte/macrophage showing three subsets. The marker genes, composition, and enrichment analysis for each subset are listed. (h) Subclustering of T cells showing

seven subsets. The marker genes and composition for each subset are listed. ***P < 0.0001, **P < 0.01, and *P < 0.05 by Student's t-test for data in (c), (d), and (e).

5) The requirement of adaptive immune cells was shown with Rag2^{-/-} mice. How ECM regulates adaptive immune cells needs to be analyzed in more depth.

Response: Thanks for the valuable suggestion. We have supplemented the in-depth description of how ECM regulates adaptive immune cells in Rag2^{-/-} mice as suggested. The related descriptions were shown in the Result and Discussion section with changes marked.

Result (Page 13-14; Line 310-317): Subclustering of MAC resulted in three subsets including PIM², AIM², and Mono² (Fig. 5g). Rag2^{-/-} samples improved the recruitment of AIM², PIM², in which AIM² owned the highest proportion (46.2%). Subclustering of T cells resulted in seven subsets (Fig. 5h). The Rag2^{-/-} groups contributed to fewer T cells in most subsets except NKT² (related to natural killer cell mediated cytotoxicity) and naive Th2², indicating the dysfunctional adaptive immune systems in Rag2^{-/-} group. The absence of suppressive Treg² might be the reason for uncontrolled type-2 AIM² accumulation and collagen deposition of fibrotic RF².

Fig. 5: Evaluation of wound healing in immunodeficient mice lacking mature T cells.

(g) Subclustering of monocyte/macrophage showing three subsets. The marker genes,

composition, and enrichment analysis for each subset are listed. (h) Subclustering of T cells showing seven subsets.

Result (Page 14; Line 322-330): To compare the scaffold-induced immune microenvironment, we also integrated marker genes of immune cells (defined in Fig. 5) with ST profiling. We confirmed the more obvious aggregation of AIM² surrounding the biomaterial in the Rag2^{-/-} sample (Fig. 6 g and h), which might contribute to the absence of the Treg² (Fig. 6 i and j).

Fig. 6: Spatial atlas of cell microenvironment around biomaterials of immunodeficient mice. (g) Spatial feature plot and violin plot showing the distribution and expression level of integrated AIM² subcluster in tissue sections. (h) Spatial feature plot and violin plot showing the distribution and expression level of Mrc1 (marker gene of AIM²) in tissue sections; IF staining of F4/80 and CD206 (white arrowheads showing the F4/80⁺CD206⁺ cells). (i) Spatial feature plot and violin plot showing the distribution and expression level of integrated Treg² subcluster in tissue sections. (j) Spatial feature plot and violin plot showing the distribution and expression level of Foxp3 (marker gene of Treg²) in tissue sections; IF staining of FOXP3 (white arrowheads showing the FOXP3⁺ cells).

Discussion (Page 16; Line 361-364): We next confirmed the requirement of T cells in skin regeneration by an immunodeficient model (Fig. 5). The absence of suppressive Treg² might be one of the reasons for the uncontrolled accumulation of type-2 AIM², which might drive collagen deposition by activating profibrotic RF².

6) Previous studies showed the roles of macrophages and gamma delta T cells are essential for hair follicle neogenesis. Previous studies showed peripheral parts of large wounds can induce hair follicle neogenesis with shh signal activation.

Relation to these previous studies is unclear.

Response: Thanks for the valuable suggestion. We have supplemented the introduction regarding the roles of macrophages and gamma delta T cells in hair follicle neogenesis, and the related descriptions were provided in the Introduction section of the manuscript with changes marked. Besides, we have supplemented the description of sonic hedgehog (shh) signaling in hair follicle neogenesis in the Result section with changes marked.

Introduction (Page 2; Line 35-48): Macrophages contribute to all phases of tissue repair, and the heterogeneity of macrophages is believed to be one of the critical orchestrators determining the healing outcome^{9,10}. Two major subpopulations, including pro-inflammatory M1-like and anti-inflammatory M2-like, have been credited with these distinct roles¹¹. It has been reported that the pro-inflammatory macrophage-elicited pro-inflammatory mediators, such as tumor necrosis factor (TNF)⁵ and interleukin-1 beta (IL-1 β)¹², effectively promote subsequent HF neogenesis. In contrast, type-2 anti-inflammatory macrophages might play an essential role in wound fibrosis by promoting fibrotic fibroblast activation and collagen cross-linking³ via fibrotic cytokines such as transforming growth factor-beta (TGF- β)¹³ and RELM α ¹⁴ or chronic phagocytosis activity⁴ at a later stage. Although the role of macrophages in pathogen clearance and tissue fibrosis¹⁵ has long been stressed, only recently have the T lymphocytes been more thoroughly investigated. In addition to $\gamma\delta$ T cells that induce HF neogenesis through the secretion of fibroblast growth factor 9 (Fgf9)⁶, the adaptive T cells are being explored to gain more understanding of its role in regulating macrophage polarization and, thus, wound regeneration².

Result (Page 5-6; Line 120-125): In contrast, enrichment analysis of ECM up-regulated genes (p value < 0.05 and $|\log_2\text{FoldChange}| > 1$) revealed enrichment of hair follicle development and mesenchyme morphogenesis driven by Wnt signaling pathway and **Hedgehog signaling pathway** in the ECM_LW group (Fig. 1i and Supplementary Fig. 2c). The role of Wnt and Hedgehog signaling pathway in regulating T cell development³⁷⁻³⁹ as well as hair follicle regeneration^{40,41} had been stressed.

Fig1i: Bulk-RNA sequencing analysis of ECM_LW versus Ctrl_LW mice on POD7 (n=3 for each group). Heatmap (left) showing hierarchical clustering of differentially expressed genes (p value < 0.05 & $|\log_2\text{FC}| > 1$) between two groups, and corresponding gene set enrichment analysis (right) showing the enriched terms in ECM_LW (top) versus Ctrl_LW (bottom) groups.

Supplementary Fig. 2c. Chord plots of enriched terms in ECM_LW group.

7) The authors need to provide sufficient introduction regarding hair follicle neogenesis and immune response in skin wounds.

Response: Thanks for the valuable suggestion. We have supplemented the introduction regarding hair follicle neogenesis immune response in skin wounds.

Introduction (Page 2-3 ; Line 33-53): The **immune system** plays a varying role in driving scar fibrosis²⁻⁴ or HF regeneration⁵⁻⁸ upon different environmental stimuli. Macrophages contribute to all phases of tissue repair, and the heterogeneity of macrophages is believed to be one of the critical orchestrators determining the healing outcome^{9,10}. Two major subpopulations, including pro-inflammatory M1-like and anti-inflammatory M2-like, have been credited with these distinct roles¹¹. It has been reported that the pro-inflammatory macrophage-elicited pro-inflammatory mediators, such as tumor necrosis factor (TNF)⁵ and interleukin-1 beta (IL-1 β)¹², effectively promote subsequent HF neogenesis. In contrast, type-2 anti-inflammatory macrophages might play an essential role in wound fibrosis by promoting fibrotic fibroblast activation and collagen cross-linking³ via fibrotic cytokines such as transforming growth factor–beta (TGF- β)¹³ and RELM α ¹⁴ or chronic phagocytosis activity⁴ at a later stage. Although the role of macrophages in pathogen clearance and tissue fibrosis¹⁵ has long been stressed, only recently have the T lymphocytes been more thoroughly investigated. In addition to $\gamma\delta$ T cells that induce HF neogenesis through the secretion of fibroblast growth factor 9 (Fgf9)⁶, the adaptive T cells are being explored to gain more understanding of its role in regulating macrophage polarization and, thus, wound regeneration². T cells coordinate the polarized immune responses through differentiation into specialized subsets of helper T cells (Th1, Th2, and Th17) and drive the type 1/2/3 paradigm of immunity¹⁶. Besides, specialized regulatory T cells (Tregs) have evolved to counterbalance the potentially detrimental

effect of the innate immune system by suppressing macrophage response¹⁷ and facilitating wound regeneration.

Response: We thank **Reviewer 2** sincerely for reviewing our manuscript and providing insightful and constructive comments. These suggestions have enabled us to produce a greatly improved manuscript whose novelty and impact are now more readily apparent.

Referee 3

Reviewer #3 (Remarks to the Author):

Yang et al present a thorough and interesting study on spatial and single-cell and spatial transcriptomic profiling of the immune landscape of skin injury after treatment with a biologic (ECM) scaffold in a mouse model. The study is of interest to a general audience within the field of wound healing and adds important state-of-the-art technological advances in characterizing immunomodulation by biomaterials. There are several considerations to be made prior to publication:

Response: We appreciate the Reviewer 3 for reviewing our manuscript and provided insightful comments, and we have carefully addressed these concerns and made a proper revision of the manuscript. Below, we address the questions and suggestions raised by Reviewer 3 point-by-point.

Major Comments

1) Standard wound healing models are usually 5mm, is there a reason why such a large defect was chosen?

Response: Thanks for the valuable suggestion. We have supplemented the reason why the large-scale defect model was chosen for the main research object in Introduction section with change marked. We also repeated the evaluation in small wound model, the related descriptions were supplemented in Result section and Fig. 7.

Introduction (Page 3; Line 59-65): Our previous studies had reported the Aligned nanofibers scaffold with an immunomodulatory effect in accelerating small skin wound (diameter = 6 mm) re-epithelialization^{22,23}. However, scar tissue induced by a 6-mm-scale wound was tiny (diameter = 1~1.5 mm). Considering the more obvious inflammatory response and larger detectable scar tissue, the large-scale full-thickness wound model (diameter = 10~20mm)²⁴⁻²⁸ provided a better media through which we

can easily evaluate the pro-regenerative effect of ECM scaffolds regarding scarless wound healing.

Reference :

- [22] Hu, C. *et al.* Dissecting the microenvironment around biosynthetic scaffolds in murine skin wound healing. *Science advances* **7**, doi:10.1126/sciadv.abf0787 (2021).
- [23] Wang, C. *et al.* The diameter factor of aligned membranes facilitates wound healing by promoting epithelialization in an immune way. *Bioactive materials* **11**, 206-217, doi:10.1016/j.bioactmat.2021.09.022 (2022).
- [24] Cui, T. *et al.* Large-Scale Fabrication of Robust Artificial Skins from a Biodegradable Sealant-Loaded Nanofiber Scaffold to Skin Tissue via Microfluidic Blow-Spinning. *Advanced materials (Deerfield Beach, Fla.)* **32**, e2000982, doi:10.1002/adma.202000982 (2020).
- [25] Yang, S. *et al.* MSC-derived sEV-loaded hyaluronan hydrogel promotes scarless skin healing by immunomodulation in a large skin wound model. *Biomedical materials (Bristol, England)* **17**, doi:10.1088/1748-605X/ac68bc (2022).
- [26] Shen, Y. *et al.* Sequential Release of Small Extracellular Vesicles from Bilayered Thiolated Alginate/Polyethylene Glycol Diacrylate Hydrogels for Scarless Wound Healing. *ACS nano* **15**, 6352-6368, doi:10.1021/acsnano.0c07714 (2021).
- [27] Zhang, Z. *et al.* Design of a biofluid-absorbing bioactive sandwich-structured Zn-Si bioceramic composite wound dressing for hair follicle regeneration and skin burn wound healing. *Bioactive materials* **6**, 1910-1920, doi:10.1016/j.bioactmat.2020.12.006 (2021).
- [28] Ahmadian, Z. *et al.* A Hydrogen-Bonded Extracellular Matrix-Mimicking Bactericidal Hydrogel with Radical Scavenging and Hemostatic Function for pH-Responsive Wound Healing Acceleration. *Advanced healthcare materials* **10**, e2001122, doi:10.1002/adhm.202001122 (2021).

Result (Page 14-15; Line 331-341) :

Biomaterials facilitate de novo HF regeneration despite wound size

To detect any differences in wound healing between small and large wounds treated with biomaterials, we repeated the experiment with small (diameter=0.6 cm) full-thickness wounds implanted with biomaterials (Fig. 7 a and b). Consistent with the ECM_LW group, the scaffold implanted in the small wound (ECM_SW) did not trigger an obvious FBR fibrous capsule and had a rapid degradation rate (Fig. 7c). Histological sections of ECM_SW tissue revealed clear signs of enhanced HF reconstruction too (Fig. 7 c and d). ST was also applied to explore the spatial

characteristics of Ctrl_SW and ECM_SW samples, and the anatomical structure was shown in Fig. 7e. We still observed the recruitment of $Cd3^+$ T cells around the implanted biomaterial, which were co-located with $Crabp1^+$ papillary fibroblasts in ECM_SW sample (Fig. 7g), confirming the pro-regenerative potential of ECM scaffolds despite of wound size.

Fig. 7: Evaluation of the healing of small full-thickness wounds treated with ECM scaffolds. (a) Workflow for evaluating skin wound healing. (b) The surgical process for skin excisional wound model of Ctrl_SW and ECM_SW group. (c) H&E staining of Ctrl_SW and ECM_SW samples. (d) Histologic quantification of de novo hair follicles on POD28 (n=5 for each group). (e) The anatomical structure of samples. (f) Spatial feature plot showing the expression of *Cd3d* (marker gene of T cells) and *Crabp1* (marker gene of papillary fibroblasts) in ST profile and corresponding

quantitative analysis. ***P < 0.0001, **P < 0.01, and *P < 0.05 by Student's t-test for data in (d).

2) For the spatial transcriptomics data, often transcriptome deviates from proteome. Immunofluorescence images of a related protein would be useful to place next to the transcriptome visualization. There is some immunohistochemistry and IF but expanding on this would strengthen the conclusions from the spatial transcriptomics.

Response: Thanks for the valuable suggestion. We have supplemented the IF staining of anti-inflammatory macrophage (F4/80⁺Cd206⁺), regulatory T cells (Foxp3⁺), Papillary fibroblast (Crabp1⁺), Reticular fibroblast (Mest⁺) as suggested (see the Fig. 1 k, n and Fig. 6 in the revised manuscript, which were helpful to strengthen the conclusions from the spatial transcriptomics. Fig. 6 below shows the example of supplemented IF stainings (marked with red box).

Result (Page 14 ; Line 322-330): Compared to WT samples, the distribution and expression level of PF² (HF-related) was reduced (Fig. 6c), which was verified by the IF staining of the corresponding marker gene Crabp1 (Fig. 6d). In contrast, the distribution and expression level of lower lineage Mest⁺ RF² were significantly increased (Fig. 6 e and f), in accordance with the healing outcome of Rag2^{-/-} groups. To compare the scaffold-induced immune microenvironment, we also integrated marker genes of immune cells (defined in Fig. 5) with ST profiling. We confirmed the more obvious aggregation of AIM² surrounding the biomaterial in the Rag2^{-/-} sample (Fig. 6 g and h), which might contribute to the absence of the Treg² (Fig. 6 i and j).

Fig. 6: (c) Spatial feature plot and violin plot showing the distribution and expression level of integrated PF² subcluster in tissue sections. (d) Spatial feature plot and violin plot showing the distribution and expression level of *Crabp1* (marker gene of PF²) in tissue sections; IF staining of CRABP1 (white arrowheads showing the CRABP1⁺ cells). (e) Spatial feature plot and violin plot showing the distribution and expression level of integrated RF² subcluster in tissue sections. (f) Spatial feature plot and violin plot showing the distribution and expression level of *Mest* (marker gene of RF²) in tissue sections; IF staining of MEST (white arrowheads showing the MEST⁺ cells). (g) Spatial feature plot and violin plot showing the distribution and expression level of integrated AIM² subcluster in tissue sections. (h) Spatial feature plot and violin plot showing the distribution and expression level of *Mrc1* (marker gene of AIM²) in tissue sections; IF staining of F4/80 and CD206 (white arrowheads showing the F4/80⁺CD206⁺ cells). (i) Spatial feature plot and violin plot showing the distribution and expression level of integrated Treg² subcluster in tissue sections. (j) Spatial feature plot and violin plot showing the distribution and expression level of *Foxp3* (marker gene of Treg²) in tissue sections; IF staining of FOXP3 (white arrowheads showing the FOXP3⁺ cells).

3) *Figure 1j – the data suggesting there are more CD3+ T cells present than CD68+ macrophages is quite contradictory to much of published literature. Confirmation with another technique such as flow cytometry would be needed (ex. CD45, CD3, CD68, F4/80)*

Response: Thanks for the valuable suggestion. We have supplemented the flow cytometry analysis to evaluate the proportion of T cells (CD45⁺CD3⁺) and macrophages (CD45⁺CD3⁻CD68⁺F4/80⁺) among live cells across all treatments as suggested (see the Fig.1j and Supplementary Fig. 3 in the revised manuscript). In accordance with previous report, the proportion of CD3⁺ T cells was lower than CD68⁺F4/80⁺ macrophages on POD7 by flow cytometry analysis. It is possible that the increased number of CD3⁺ cells over CD68⁺ cells in the IHC analysis results was due to background staining. We apologize for the previous mistake, which may have made some of this evidence unclear. The re-calculation results will be presented in the next response of the comment.

Fig. 1j: Proportions of T cells (CD45⁺CD3⁺) and macrophages (CD45⁺ CD3⁻ F4/80⁺CD68⁺) cell populations in the wound environment on POD7, determined by flow cytometry (% = the number of target cells / the number of all single live cells) (n=3 for each group).

Supplementary Fig. 3: Gating scheme for T cells and macrophages. Multicolor flow cytometry gating strategy to isolate T cells ($CD45^+CD3^+$) and macrophages ($CD45^+CD3^-CD68^+F4/80^+$) from single cell suspension of Ctrl_LW (a) and ECM_LW (b) samples on POD7.

• *Fig 1j – it looks like there is more CD68 staining in the ECM scaffold group but the quantification states otherwise. Is this background staining? Please provide primary delete (secondary only) controls for these stains.*

Response: Thanks for the valuable suggestion. It is possible that the increased number of CD3⁺ cells over CD68⁺ cells in the IHC analysis results was due to

background staining. We apologize for the previous mistake, which may have made some of this evidence unclear. We have supplemented the primary delete (secondary only) controls for these stains and re-calculated the number of CD3⁺ and CD68⁺ cells of both groups as suggested (see the Supplementary Fig.2d).

Supplementary Fig.2d: IHC staining of CD3 and CD68 on POD7 and corresponding quantitative analysis (n=5 for each group).

Minor comments

1) Line 66 – what does LW stand for? Large wound? Please state in explicitly in text

Response: Thanks for the valuable suggestion. LW stands for ‘Large wound’. We have checked it carefully and supplemented the detailed elaborations in text as suggested. The related descriptions were provided in the Result section of the manuscript with changes marked.

Result (Page 4 ; Line 82-85): We placed ECM scaffolds below the large wound (diameter = 1.5 cm) in the **ECM_LW (Large wound treated with ECM scaffold)** group, while the **Ctrl_LW (Large wound treated with saline)** group received no biomaterials (Fig. 1b).

2) Subject-verb agreement and other small grammar items should be double-checked

Response: Thanks for the valuable suggestion. We have sent professional language editing service of Elsevier and double-checked the grammar.

3) Line 73 – what does POD stand for? Please state explicitly in text

Response: Thanks for the valuable suggestion. POD stands for ‘postoperative day’. We have checked it carefully and supplemented the full name in the text as suggested (see the text in line 95-96 with change marked).

4) It is hard to evaluate the histology images, please provide high-resolution histology

Response: Thanks for the valuable suggestion. We have checked it carefully and supplemented the high-resolution histology as suggested (see the Figures in the revised manuscript).

5) Fig 3 – c,d,g,h need labels on Control v ECM scaffold (as in panel a)

Response: Thanks for the valuable suggestion. We have checked it carefully and supplemented the labels of Ctrl_LW & ECM_LW group as suggested (see the Fig.3 d, e, i, l, and m in the revised manuscript). Fig. 3 below shows the example of supplemented labels (marked with red box).

Fig.3: (d) Spatial feature plot showing the distribution of IFE^B and IFE^D subclusters in tissue sections. (e) Spatial feature plot showing the distribution of PF¹ and RF¹ subclusters in tissue sections. (i) Spatial feature plot showing the distribution of MAC subclusters in tissue sections. (l) Spatial feature plot and violin plot showing TC¹ distribution and expression level in tissue sections. (m) Spatial feature plot and violin plot showing the distribution and expression level of Treg¹ in tissue sections. =

6) Fig 5 – histopathology images (g) are very small, would suggest moving panel h to supplement (one would expect fewer T cells in a Rag^{-/-} mouse, the confirmation is good for supplemental) and increasing the size of panel g.

Response: Thanks for the valuable suggestion. We have checked it carefully and moving panel h to supplementary figure as suggested (see the Supplementary Fig. 7 in the revised manuscript).

Supplementary Fig. 7: Histological analysis of WT and Rag2^{-/-} mice treated with biomaterials. (a) IHC staining of CD3, CD68, and Ly6G. (b) semi-quantification of CD3⁺ T cell infiltration (n=3 for each group).

7) Fig5i – it would not be number (No.) of cells, but Fraction of cells since you are reporting in %

Response: Thanks for the valuable suggestion. We have checked it carefully and revised the caption into ‘Fraction of cells (%)’ (see the Fig. 5f in the revised manuscript). Fig. 5f below shows the revised caption (marked with red box).

Fig. 5f: Subclustering of fibroblasts and fibroblast-like cells showing two fibroblast-like subsets and five fibroblast subsets. Marker genes for fibroblast subsets are listed.

Response: We appreciate **Reviewer 3** for reviewing our manuscript and providing insightful comments, and these suggestions have enabled us to provide a highly improved manuscript.

Once again, We would like to express our gratitude to the reviewers for recognizing our work and offering constructive suggestions. The six-month revision period has dramatically enhanced the quality of our research and broadened our understanding, which inspired us to conduct more in-depth studies in future works.

REVIEWERS' COMMENTS

Reviewer #1 (Remarks to the Author):

The authors have addressed all my comments, I have no further comments.

Reviewer #2 (Remarks to the Author):

The authors rigorously addressed my previous comments. I have no further comments.

Reviewer #3 (Remarks to the Author):

The authors have adequately addressed all scientific comments, and those that remain are technical details to support the repeatability and validity of findings.

>Supplementary Figure 3: PE is a very bright fluorophore and CD3 a robust antigen. One would expect brighter signal than what is displayed in the gating scheme. Please display the fluorescence minus on control of this antibody to confirm that the gate is placed in the appropriate location and that the population is true and not contaminated with any debris. The same for F4/80. Is the CD45^{lo} population background or is that staining? FMO controls will support these findings.

>The IHC still appears to have some background staining, especially on the ECM itself. It is possible to optimize staining to minimize this background and allow for clearer evaluation of cells. It is my concern that there might still be some under/over counting due to the inability to detect cells in the ECM due to this background.

>Individual data points should be displayed and information on the quantification should be described (ex. histology quantification is a result of how many fields from how many mice)

Replies to the Reviewers' Comments

Referee 1

Reviewer #1 (Remarks to the Author):

The authors have addressed all my comments, I have no further comments.

Response: We thank Reviewer 1 for reviewing our manuscript and providing insightful and constructive comments so that we can improve it further.

Referee 2

Reviewer #2 (Remarks to the Author):

The authors rigorously addressed my previous comments. I have no further comments.

Response: We appreciate Reviewer 2 for carefully reading our manuscript and making insightful, critical, and constructive feedback, which has enabled us to prepare a greatly improved manuscript.

Referee 3

Reviewer #3 (Remarks to the Author):

The authors have adequately addressed all scientific comments, and those that remain are technical details to support the repeatability and validity of findings.

Response: We appreciate Reviewer 3 for reviewing our manuscript and providing insightful comments, we have carefully addressed these concerns and made a proper revision of the manuscript. Below, we address the questions and suggestions raised by Reviewer 3 point-by-point.

>Supplementary Figure 3: PE is a very bright fluorophore and CD3 a robust antigen. One would expect brighter signal than what is displayed in the gating scheme. Please display the fluorescence minus on control of this antibody to confirm that the gate is placed in the appropriate location and that the population is true and not contaminated with any debris. The same for F4/80. Is the CD45^{lo} population background or is that staining? FMO controls will support these findings.

Response: Thanks for the valuable suggestion. We have supplemented the fluorescence minus one (FMO) control of CD3, F4/80, CD45, and CD68 to support the gating scheme for T cells and macrophages. The related images were provided in Supplementary Fig. 3 (in yellow boxes) of Supplementary information, which was helpful in strengthening the conclusions from the flow cytometry.

Supplementary Figure 3. Gating scheme for T cells and macrophages. Multicolor flow cytometry gating strategy to isolate T cells ($CD45^+CD3^+$) and macrophages ($CD45^+CD3^-CD68^+F4/80^+$) from single cell suspension. Representative flow cytometry plots of Ctrl_LW (a) and ECM_LW (b) samples on POD7. Abbreviation: FMO, Fluorescence Minus One.

>The IHC still appears to have some background staining, especially on the ECM itself. It is possible to optimize staining to minimize this background and allow for clearer evaluation of cells. It is my concern that there might still be some

under/over counting due to the inability to detect cells in the ECM due to this background.

Response: Thanks for the valuable suggestion. We have optimized the staining of IHC to minimize this background and allow for a clearer evaluation of cells. The related images were provided in Supplementary Fig. 2d and 7a of Supplementary information, which was helpful in strengthening the conclusions from the flow cytometry.

Supplementary Figure 2. Analysis of Ctrl_LW and ECM_LW samples. (d)

Representative IHC images of stained T cells (CD3⁺) and monocyte-macrophages (CD68⁺) on POD7 and corresponding quantitative analysis (Data are presented as mean ± SD, n=5 biologically independent samples, two-tailed t-test, CD3 ***p* = 0.007; CD68 ***p* = 0.008). *p* value: **p* < 0.05, ***p* < 0.01, ****p* < 0.001, and *****p* < 0.0001.

Supplementary Figure 7. Histological analysis of WT and Rag2^{-/-} mice treated with biomaterials. (a) IHC staining of CD3, CD68, and Ly6G. (b) semi-quantification of CD3⁺ T cell infiltration (Data are presented as mean ± SD , n=5

biologically independent samples, two-tailed t-test, $***p = 0.000210$). p value: $*p < 0.05$, $**p < 0.01$, $***p < 0.001$, and $****p < 0.0001$.

>Individual data points should be displayed and information on the quantification should be described (ex. histology quantification is a result of how many fields from how many mice)

Response: Thanks for the valuable suggestion. We have checked and overlaid the corresponding data points (as dot plots) accompanied by precise n numbers in Fig. 1j, 5c, 5d, and Supplementary Fig. 2d, 7b as suggested. The quantification information of images has been supplemented in “Statistics and Reproducibility” in the methods section.

Fig. 1 Evaluation of the wound healing process treated with ECM scaffolds. (j) Proportions of T cells (CD45⁺CD3⁺) and macrophages (CD45⁺ CD3⁺F4/80⁺CD68⁺) cell populations in the wound environment on POD7, determined by flow cytometry (% = the number of target cells / the number of all single live cells) (Data are presented as mean \pm SD, n=3 biologically independent samples, two-tailed t-test, T cells $*p = 0.013$; Macrophages $*p = 0.016$). p value: $*p < 0.05$, $**p < 0.01$, $***p < 0.001$, and $****p < 0.0001$.

Fig. 5 Evaluation of wound healing in immunodeficient mice lacking mature T cells. (c) Residual defect area on POD 7 (Data are presented as mean \pm SD, n=4 biologically independent samples, two-tailed t-test, * $p = 0.014$). (d) Semiquantitative evaluation of gap width (Data are presented as mean \pm SD, n=4 biologically independent samples, two-tailed t-test, *** $p = 0.00046$). p value: * $p < 0.05$, ** $p < 0.01$, *** $p < 0.001$, and **** $p < 0.0001$.

Supplementary Figure 2. Analysis of Ctrl_LW and ECM_LW samples. (d) Representative IHC images of stained T cells (CD3⁺) and monocyte-macrophages (CD68⁺) on POD7 and corresponding quantitative analysis (Data are presented as mean \pm SD, n=5 biologically independent samples, two-tailed t-test, CD3 ** $p = 0.007$; CD68 ** $p = 0.008$). p value: * $p < 0.05$, ** $p < 0.01$, *** $p < 0.001$, and **** $p < 0.0001$.

Supplementary Figure 7. Histological analysis of WT and Rag2^{-/-} mice treated with biomaterials. (a) IHC staining of CD3, CD68, and Ly6G. (b) semi-quantification of CD3⁺ T cell infiltration (Data are presented as mean \pm SD , n=5 biologically independent samples, two-tailed t-test, *** $p = 0.000210$). p value: * $p < 0.05$, ** $p < 0.01$, *** $p < 0.001$, and **** $p < 0.0001$.

Statistics and reproducibility

The precise sample number for each experiment was indicated in the figure legends. For Fig. 1g, 3k, 3n, 6d, 6f, 6h, and 6j, representative images were shown from one of three biological repeats; For Fig. 5b and Supplementary Fig. 2a, representative images were shown from one of four biological repeats; For Fig. 1e, 7c and Supplementary Fig. 1c, 2d, 7a, representative images were shown from one of five biological repeats.

Once again, We would like to express our gratitude to the reviewers for recognizing our work and offering constructive suggestions. Your insightful comments have dramatically enhanced the quality of our research and broadened our understanding, which inspired us to conduct more in-depth studies in future works.